# MIND bomb 2 prevents RIPK1 kinase activity-dependent and -independent apoptosis through ubiquitylation of cFLIP$_L$

Osamu Nakabayashi [1], Hirotaka Takahashi[2], Kenta Moriwaki[1], Sachiko Komazawa-Sakon[1], Fumiaki Ohtake[3], Shin Murai [1], Yuichi Tsuchiya[1], Yuki Koyahara[1], Yasushi Saeki [4], Yukiko Yoshida[4], Soh Yamazaki[1], Fuminori Tokunaga [5], Tatsuya Sawasaki [2] & Hiroyasu Nakano [1✉]

Mind bomb 2 (MIB2) is an E3 ligase involved in Notch signalling and attenuates TNF-induced apoptosis through ubiquitylation of receptor-interacting protein kinase 1 (RIPK1) and cylindromatosis. Here we show that MIB2 bound and conjugated K48– and K63–linked polyubiquitin chains to a long-form of cellular FLICE-inhibitory protein (cFLIP$_L$), a catalytically inactive homologue of caspase 8. Deletion of *MIB2* did not impair the TNF-induced complex I formation that mediates NF-κB activation but significantly enhanced formation of cytosolic death-inducing signalling complex II. TNF-induced RIPK1 Ser$^{166}$ phosphorylation, a hallmark of RIPK1 death-inducing activity, was enhanced in *MIB2* knockout cells, as was RIPK1 kinase activity-dependent and -independent apoptosis. Moreover, RIPK1 kinase activity-independent apoptosis was induced in cells expressing cFLIP$_L$ mutants lacking MIB2-dependent ubiquitylation. Together, these results suggest that MIB2 suppresses both RIPK1 kinase activity-dependent and -independent apoptosis, through suppression of RIPK1 kinase activity and ubiquitylation of cFLIP$_L$, respectively.

---

[1] Department of Biochemistry, Toho University School of Medicine, 5-21-16 Omori-Nishi, Ota-ku, Tokyo 143-8540, Japan. [2] Division of Cell-Free Sciences, Proteo-Science Center (PROS), 3 Bunkyo-cho, Matsuyama, Ehime 790-8577, Japan. [3] Institute for Advanced Life Sciences, Hoshi University, 2-4-41 Ebara, Shinagawa-ku, Tokyo 142-8501, Japan. [4] Laboratory of Protein Metabolism, Tokyo Metropolitan Institute of Medical Science, 2-1-6 Kamikitazawa, Setagaya-ku, Tokyo 156-8506, Japan. [5] Department of Pathobiochemistry, Graduate School of Medicine, Osaka City University, 1-4-3 Asahi-machi, Abeno-ku, Osaka 545-8585, Japan. ✉email: hiroyasu.nakano@med.toho-u.ac.jp

The mind bomb (MIB) family of proteins comprises MIB1 and MIB2, which are considered to be involved in the Notch signalling pathway[1–3]. MIB1 and MIB2 are E3 ligases and ubiquitylate a variety of cellular proteins, including Delta and Jagged. Conditional deletion of *Mib1* results in defects in the Notch signalling pathways, such as defects in arterial specification, cerebellum and skin development, syndactylism, and T- and B-cell development[4,5]. Although MIB2 is reported to be involved in myoblast fusion, hippocampus-dependent memory formation, and NF-κB activation[1,6,7], *Mib2-/-* mice do not exhibit apparent defects, with the exception of exencephaly in some[3]. However, the function of MIB2 in the tumour necrosis factor (TNF)-induced signalling pathway remains unknown. Recent studies, including ours, have reported that MIB2 ubiquitylates receptor-interacting protein kinase (RIPK) 1 and cylindromatosis (CYLD), attenuating TNF-induced apoptosis[8,9].

TNF is a multifunctional cytokine and involved in various biological responses, including inflammation, organ development, cell adhesion, and cell death[10]. TNF induces complex I, the multiprotein complex composed of TNF receptor-associated death domain (TRADD), TNF receptor-associated factor (TRAF)2, RIPK1, and cellular inhibitor of apoptosis (cIAP)1 and 2[11,12]. Recruited cIAP1 and 2 conjugate K63–linked polyubiquitin chains to themselves and to RIPK1, which then act as platforms to recruit the linear ubiquitin chain assembly complex (LUBAC) and TGFβ-activated kinase (TAK)1, respectively. Recruited LUBAC conjugates linear-type ubiquitin chains to NF-κB essential modulator (NEMO) and RIPK1, which then further recruits the IκB kinase (IKK) complex, resulting in NF-κB activation. Activation of NF-κB promotes cell survival and induces the expression of inflammatory cytokine genes. In contrast, when cells are treated with IAP inhibitors and TAK1 inhibitors that block ubiquitylation and phosphorylation of RIPK1, respectively, RIPK1 undergoes autophosphorylation. Phosphorylated RIPK1 provides a platform to form the death-inducing signalling complex, which is referred to as complex IIb or ripoptosome[13,14]. Moreover, Tank binding kinase (TBK)1 is recruited to the complex I and phosphorylates RIPK1, thereby attenuating TNF-induced cell death[15]. Complex IIb is composed of RIPK1, FADD, and caspase 8[16]. In the presence of protein synthesis inhibitor cycloheximide (CHX), treatment of cells with TNF induces complex IIa, which is composed of TRADD, FADD, and caspase 8. Multimerization of caspase 8 is induced in the complex and results in autoactivation of the caspase. Although complex I may evolve into complex IIa or IIb under certain conditions, the detailed molecular switches are not fully understood[12,13].

Cellular FLICE-inhibitory protein (cFLIP) is a caspase 8 homologue that suppresses death-receptor mediated apoptosis[17,18]. cFLIP is encoded by *CFLAR*; alternative splicing results in two forms, the long form (cFLIP$_L$) and the short form (cFLIPs)[18]. We and others have reported that tissue-specific deletion of *Cflar* results in severe inflammation and tissue remodelling in various tissues due to enhanced cell death[18]. cFLIP is an unstable protein and degraded by the proteasome/ubiquitin-dependent pathway[19,20]. Although ITCH and LUBAC have been identified as E3 ubiquitin ligases for cFLIP$_L$[20,21], whether E3 ubiquitin ligases other than ITCH or LUBAC are also involved in the regulation of cFLIP$_L$ is unclear. As cFLIP$_L$ plays a crucial role in preventing death receptor-induced apoptosis, manipulation of the expression or function of cFLIP$_L$ may provide a means of treating cancer or other diseases by modulating cell death.

## Results

### Identification of MIB2 as a cFLIP$_L$-interacting E3 ubiquitin ligase. To identify a novel E3 ubiquitin ligase that ubiquitylates

cFLIP$_L$, we used a wheat, cell-free-based, protein-protein interaction detection system[22]. We generated 223 biotinylated RING-type E3 ligases and FLAG-tagged cFLIP$_L$ in vitro and tested their interaction (Fig. 1a). We focused on seven E3 ubiquitin ligases with high binding scores for cFLIP$_L$ (Supplementary Data 1, Fig. 1b). To test whether these seven E3 ligases ubiquitylate cFLIP$_L$, we transiently transfected HEK293 cells with expression vectors for these ligases in addition to cFLIP$_L$ and HA-ubiquitin. MIB2 efficiently ubiquitylated cFLIP$_L$, whereas TRIM31 and cIAP2 weakly ubiquitylated cFLIP$_L$ (Fig. 1c). A previous study reported that ITCH also ubiquitylates cFLIP$_L$[20]; therefore, we compared the ability of MIB2 and ITCH to ubiquitylate cFLIP$_L$. Although MIB2 and, to a lesser extent, ITCH ubiquitylated cFLIP$_L$, neither of them ubiquitylated cFLIPs (Fig. 1d). ITCH, but not MIB2, weakly ubiquitylated a protease-inactive mutant caspase 8 (caspase 8 C/S), suggesting that MIB2 is a relatively specific ligase for cFLIP$_L$. Of note, because overexpression of caspase 8 rapidly induced apoptosis, we used a protease-inactive mutant of caspase 8 (caspase 8 C/S) that harboured mutation of cysteine at 360 to serine in transient transfection assay. To test the interaction between cFLIP$_L$ and MIB2, we transiently transfected HEK293 cells with the indicated expression vectors, and cell lysates were immunoprecipitated with anti-Myc antibody. Co-immunoprecipitated proteins were detected by anti-FLAG antibody. We found that cFLIP$_L$, but not ITCH, interacted with MIB2 (Fig. 1e). Endogenous MIB2 was also immunoprecipitated with anti-cFLIP antibody (Fig. 1f). In addition, the immunoprecipitates blotted with anti-cFLIP antibody exhibited multiple bands (Fig. 1f). To verify the specificity of these bands, we knocked down cFLIP proteins in HeLa cells treated with *CFLAR* siRNA. These multiple bands disappeared in the immunoprecipitates from HeLa cells treated with *CFLAR* siRNA (Supplementary Fig. 1a, b), suggesting that they were indeed cFLIP$_L$. Given that the middle band was identical to the expected molecular size of cFLIP$_L$, we assumed that the upper and lower bands were modified and degraded cFLIP$_L$, respectively.

### Delineation of the cFLIP$_L$ domain responsible for interaction with MIB2. Next, we tested whether MIB2 interacts with cFLIPs or caspase 8 C/S. Co-transfection experiments revealed that MIB2 interacted with full-length cFLIP$_L$ and the caspase-like domain (ΔN193–480) of cFLIP$_L$, but not with cFLIPs or caspase 8 C/S (Fig. 2a). Consistent with this result, cFLIP$_L$ formed dot-like structures that completely overlapped with MIB2 in the cells (upper panels, Supplementary Fig. 2a, b). In contrast, caspase 8 C/S was diffusely distributed, whereas cFLIPs formed filamentous structures (middle and lower panels, Supplementary Fig. 2a, b). None of these structures colocalized with the dot-like structures of MIB2.

To further delineate the MIB2-binding domain of cFLIP$_L$, we generated a series of cFLIP$_L$ deletion mutants and tested their binding to MIB2 (Fig. 2b). A deletion mutant containing amino acids 1–260 interacted with MIB2, but the mutant containing 1–192 did not (Fig. 2c), suggesting that amino acids 193–260 are required for the binding of cFLIP$_L$ to MIB2. This region, when fused to glutathione S-transferase (GST) (G193–260), interacted with MIB2, but a mutant lacking the region (Δ193–260) did not (Fig. 2d). Subsequent analysis revealed that the N-terminal portion of the caspase-like domain of cFLIP$_L$ fused to GST (G239–260) still interacted with MIB2 (Fig. 2e–f). Hereafter, we refer to this region as a docking site for MIB2 (DM domain). We found that the last three C-terminal amino acids of the DM domain are aspartic acid, cysteine, and isoleucine (DCI), and are conserved in human and murine cFLIP$_L$, but not in human or murine caspase 8 (Fig. 2g). Intriguingly, a mutant without DCI

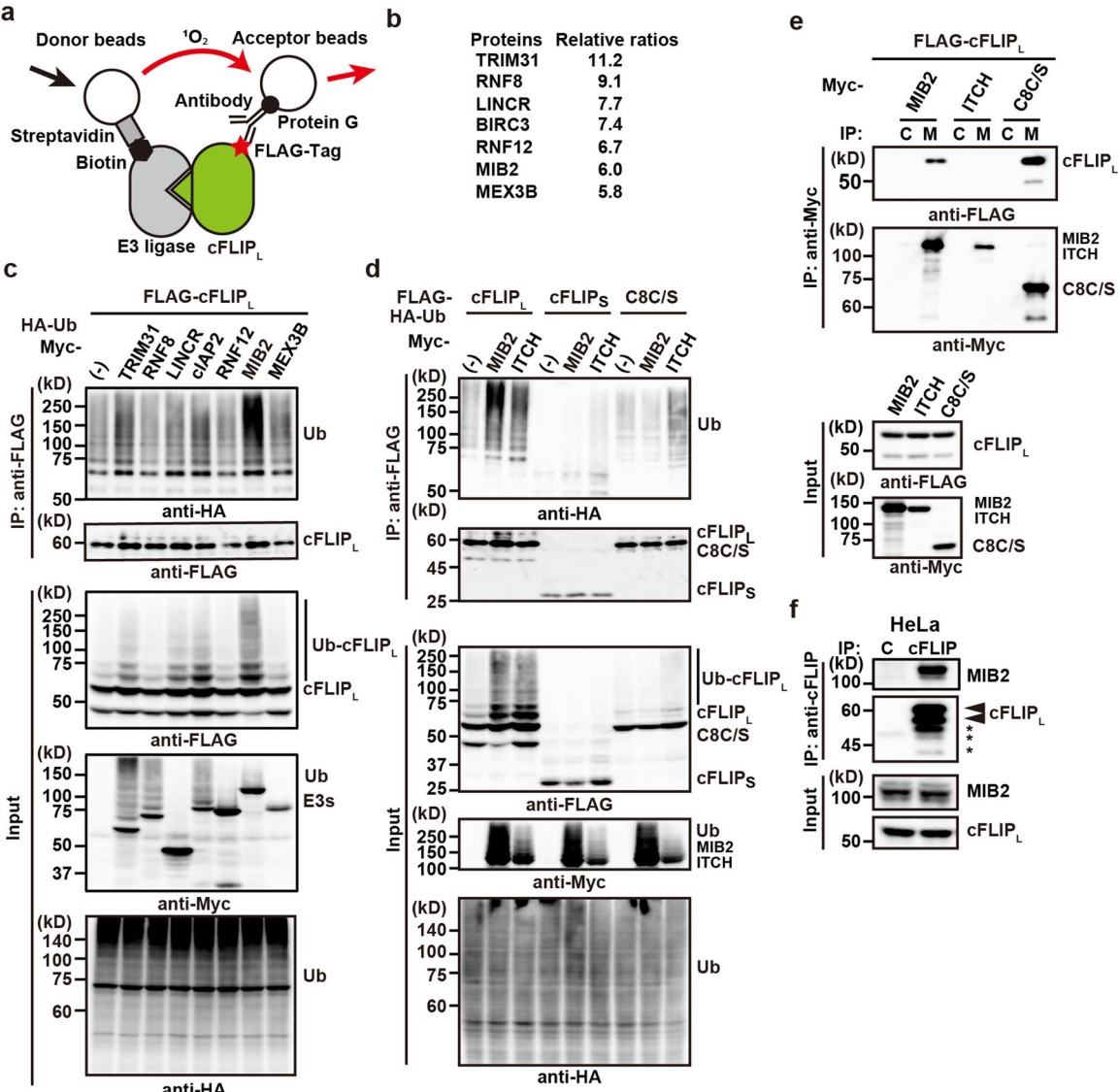

**Fig. 1 Identification of MIB2 as a cFLIP$_L$-interacting E3 ubiquitin ligase. a** Strategy for identifying cFLIP$_L$-interacting E3 ubiquitin ligase by AlphaScreening. When biotinylated E3 ligase interacts with FLAG-tagged cFLIP$_L$, singlet oxygen generated from donor beads is transferred to acceptor beads, resulting in emission. **b** The top seven E3 ligases exhibiting high binding to cFLIP$_L$. Relative ratios were calculated by dividing the relative intensity of binding to cFLIP$_L$ for each E3 ligase by the relative intensity of binding to dihydrofolate reductase. Results are the average ratios of two independent experiments. **c** HEK293 cells were transfected with the indicated E3 ligases along with FLAG-cFLIP$_L$ and HA-Ub. After immunoprecipitation with anti-FLAG antibody, ubiquitylation and total amounts of immunoprecipitated cFLIP$_L$ were analysed by immunoblotting with the indicated antibodies. Expression of transfected proteins and ubiquitylated proteins were verified by the indicated antibodies using cell lysates. **d** HEK293 cells were transfected and analysed as in (**c**). Because transfection of caspase 8 rapidly induced cell death, we used a protease-inactive mutant of caspase 8 (C8C/S). **e** HEK293 cells were transfected with the indicated expression vectors. Cell lysates were immunoprecipitated with control Ig (C) or anti-Myc (M) antibodies, and co-immunoprecipitated and immunoprecipitated proteins were analysed by immunoblotting with anti-FLAG and anti-Myc antibodies, respectively. Protein expression was verified in cell lysates using the indicated antibodies. **f** Endogenous cFLIP$_L$ interacts with MIB2. HeLa cells were immunoprecipitated with control Ig (C) or anti-cFLIP antibody. Co-immunoprecipitated MIB2 was analysed by immunoblotting with anti-MIB2 antibody. Immunoprecipitated cFLIP$_L$ was analysed by anti-cFLIP antibody. The upper and lower arrowheads indicate modified and unmodified cFLIP$_L$, respectively. Asterisks indicate degraded bands of cFLIP$_L$. Protein expression was verified by the indicated antibodies. Results are representative of two independent experiments (**c**–**f**).

(G239–257) lacked the ability to bind MIB2 (Fig. 2f–g), and DCI or DCV sequences are highly conserved among various species (Fig. 2h).

**MIB2 ubiquitylates cFLIP$_L$ in a RING domain-dependent manner**. MIB2 is composed of two MIB and zinc finger (ZF) domains at the N-terminus, nine ankyrin repeats at the centre, and two RING finger domains at the C-terminus (Fig. 3a). We hypothesised that the two RING finger domains of MIB2 might

be responsible for ubiquitylation of cFLIP$_L$. To test this hypothesis, we generated MIB2 mutants in which cysteine(s) to serine(s) mutations were introduced in the first RING (C900S, R1M), second RING (C977S, R2M), or both RING domains (CC900/977SS, R1R2M), and a MIB2 mutant lacking both RING domains (ΔR1R2). R2M, R1R2M, and ΔR1R2, but not wild-type (WT) or R1M, lost the ability to ubiquitylate cFLIP$_L$ (Fig. 3b). The results indicate that the second RING domain is required for ubiquitylation of cFLIP$_L$.

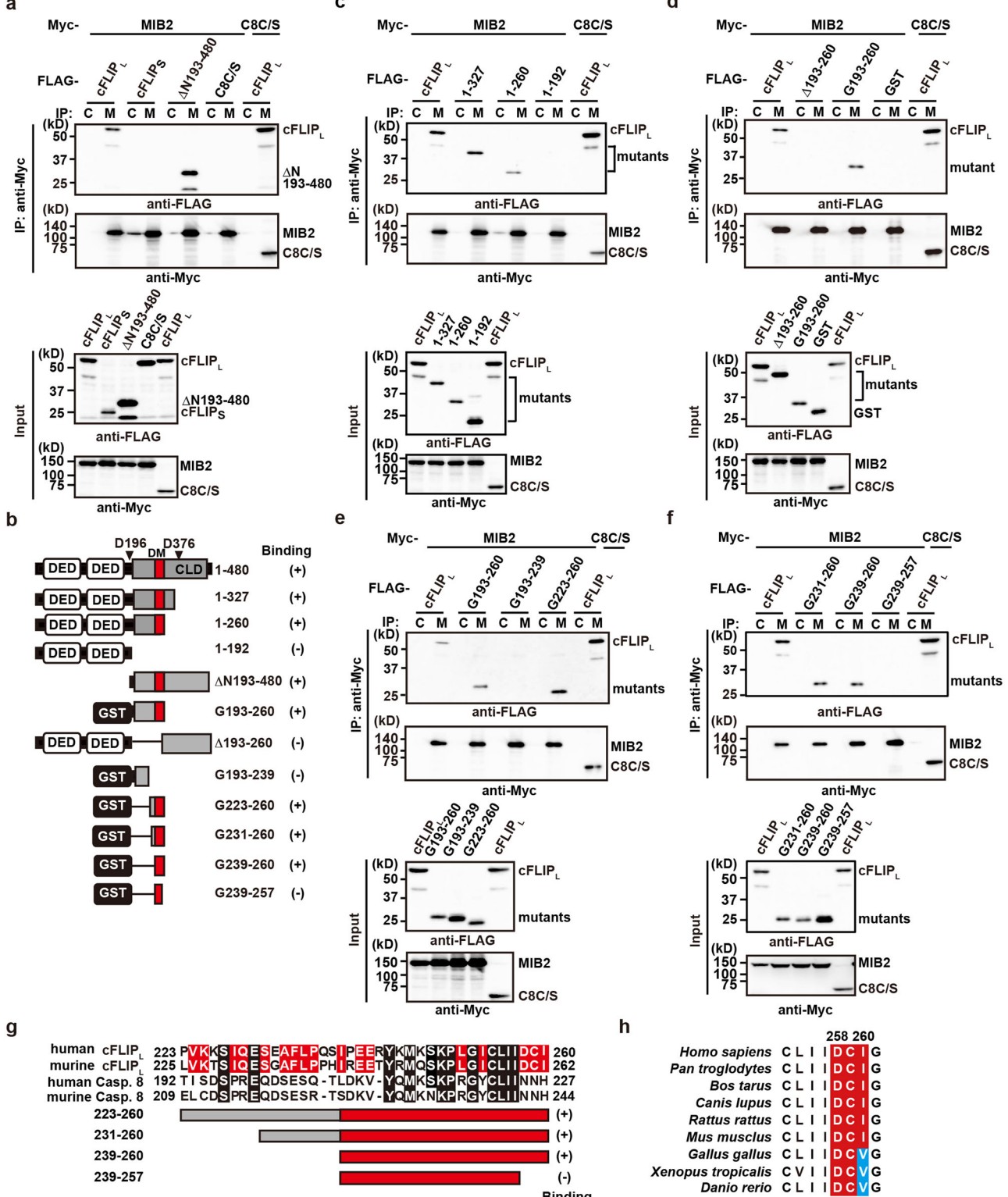

**Fig. 2 Delineation of the cFLIP_L domain responsible for interaction with MIB2. a, c–f** HEK293 cells were transfected with the indicated expression vectors and analysed as in Fig. 1e. **b** Diagrams of cFLIP_L deletion mutants and GST-fusion proteins containing the indicated amino acids of cFLIP_L, and a summary of their binding to MIB2 using co-transfection experiments. C8C/S, a protease-inactive mutant of caspase 8; DED, death effector domain; CLD, caspase-like domain; GST, glutathione S-transferase. D196 and D376 indicate the location of aspartic acid residues cleaved by activated caspase 8. All results are representative of two independent experiments. **g** Amino acid alignment of human and murine cFLIP_L and caspase 8. GST-cFLIP_L mutants containing the corresponding amino acids and their MIB2 binding ability are shown. Red boxes indicate a docking site for MIB2 (DM). **h** Alignment of the DCI sequences of cFLIP_L among various species. Red and blue boxes highlight identical and similar amino acids to the DCI sequence, respectively. Each number indicates the position of the amino acids of human cFLIP_L.

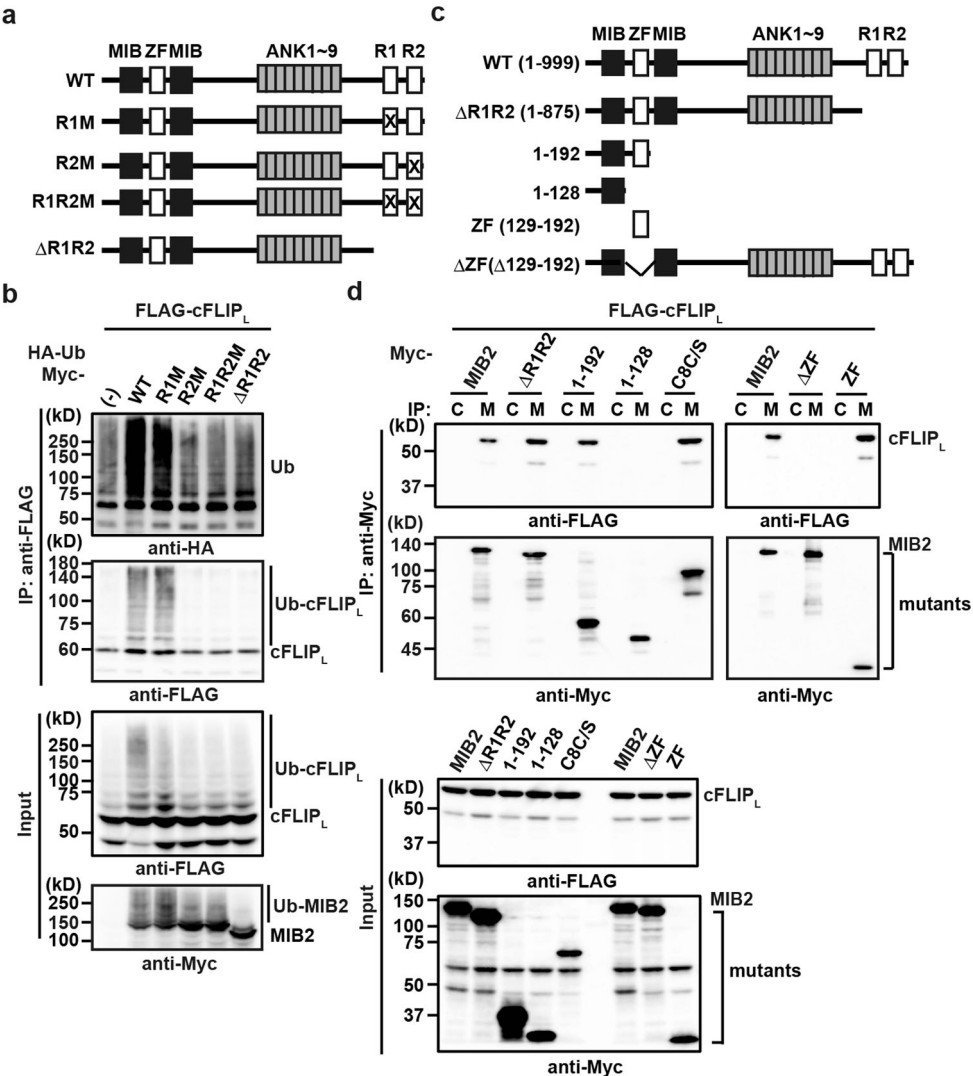

**Fig. 3 MIB2 ubiquitylates cFLIP_L in a RING domain-dependent manner. a** Domain structures of MIB2 and their mutants. ZF, zinc finger; ANK, ankyrin repeats; R, RING finger domain. X indicates the RING finger mutants in which active cysteines at 900 and 977 are replaced by serines, including R1M (C900S), R2M (C977S), and R1MR2M (CC900/977SS). **b** HEK293 cells were transfected and analysed as Fig. 1c. **c** Deletion mutants of MIB2. Numbers indicate the amino acids included in the respective mutant. **d** HEK293 cells were transfected and analysed as in Fig. 1e. C8C/S indicates a protease-inactive mutant of caspase 8. Results are representative of two independent experiments (**b, d**).

Next, we determined the cFLIP_L binding region on MIB2. We generated a series of MIB2 deletion mutants and co-transfected them into HEK293 cells with cFLIP_L (Fig. 3c). Co-immunoprecipitation revealed that a MIB2 mutant lacking the N-terminal ZF domain (ΔZF) did not interact with cFLIP_L, and that the N-terminal ZF domain alone was sufficient for binding between MIB2 and cFLIP_L (Fig. 3d).

**MIB2 conjugates K48- and K63-linked polyubiquitin chains to cFLIP_L.** Accumulating evidence indicates that different types of polyubiquitin chains have unique functions[23,24]. For example, K48- and K63-linked polyubiquitin chains are involved in protein degradation and signal transduction, respectively. To determine which types of polyubiquitin chains are generated on cFLIP_L by MIB2, we transfected HEK293 cells with single lysine to arginine (KR) mutants of ubiquitin, as well as cFLIP_L in the absence or presence of MIB2. As shown in Fig. 4a, expression of K48R and K63R ubiquitin mutants substantially reduced MIB2-dependent polyubiquitylation of cFLIP_L. Consistently, K48- and K63-linked

polyubiquitin-specific antibodies recognised K48- and K63-specific polyubiquitin chains anchored to cFLIP_L in the presence of MIB2 (Fig. 4b). In HeLa cells treated with or without TNF for 8 h, we immunoprecipitated endogenous cFLIP using anti-cFLIP antibody, followed by detection with specific antibodies against anti-K48- or anti-K63-linked polyubiquitin chains. K48- and K63-linked ubiquitylation of cFLIP_L was already present in untreated HeLa cells and did not change with TNF stimulation (Fig. 4c).

To address whether MIB2 is primarily responsible for ubiquitylation of cFLIP_L, we generated *MIB2* KO HeLa cells using the *CRISPR-Cas9* method. We found that the K63–linked polyubiquitin chains of cFLIP_L and, to a lesser extent, the K48-linked polyubiquitin chains were greatly reduced in *MIB2* KO HeLa cells compared with WT HeLa cells (Fig. 4d). To confirm the results obtained using K48- and K63-linked polyubiquitin specific antibodies, we quantified the polyubiquitin chains by mass spectrometry using immunoprecipitates obtained with anti-cFLIP antibody in cell lysates from untreated WT and *MIB2* KO HeLa cells. Most of the ubiquitylated chains on cFLIP_L were K48– and K63-linked, and some were K6- and K11-linked. These

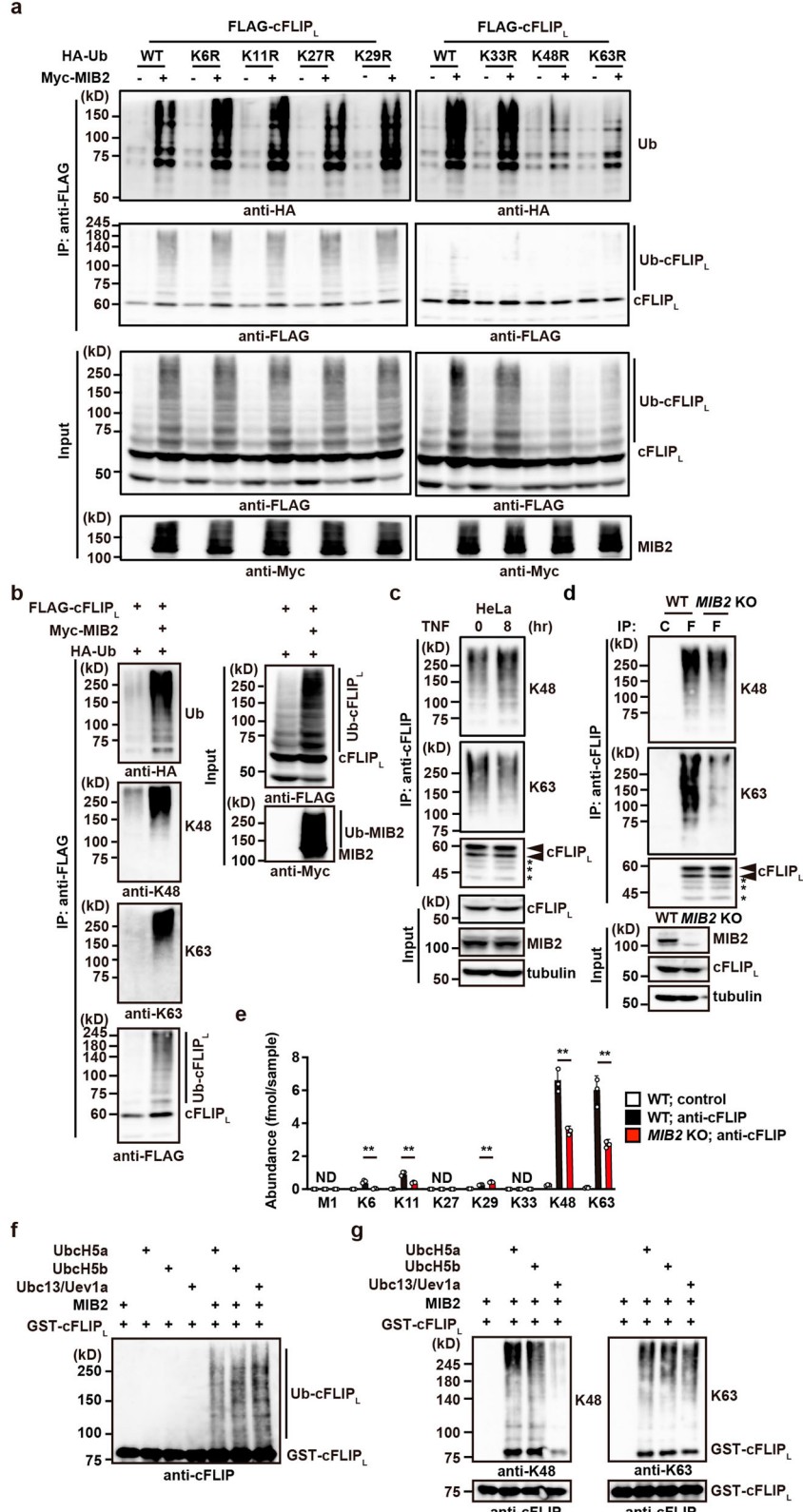

chains were reduced by half in *MIB2* KO HeLa cells compared to WT HeLa cells (Fig. 4e). Under our experimental conditions, we did not detect M1-linked polyubiquitin chains.

We verified that MIB2 ubiquitylated cFLIP$_L$ by an in vitro ubiquitylation assay using GST-cFLIP$_L$ as a substrate and immunoprecipitated MIB2 from transfected HEK293 cells as an E3 ligase (Fig. 4f). We found that MIB2 attached both K48-

and K63-linked polyubiquitin chains to cFLIP$_L$ when UbcH5a or UbcH5b were used as ubiquitin-conjugating enzymes. In contrast, MIB2 attached only K63-linked polyubiquitin chains to cFLIP$_L$ when Ubc13 and Uev1a were used as ubiquitin-conjugating enzymes (Fig. 4g). These results indicate that MIB2 mainly anchors K48- and K63-linked polyubiquitin chains to cFLIP$_L$.

**Fig. 4 MIB2 conjugates K48- and K63-linked polyubiquitin chains to cFLIP_L. a** HEK293 cells were transfected with FLAG-cFLIP_L and HA-Ub (wild-type or single lysine to arginine mutant at the indicated position) in the absence or presence of Myc-MIB2. Ubiquitylated cFLIP_L was analysed as in Fig. 1c. Protein expression was verified by immunoblotting with the indicated antibodies. **b** HEK293 cells were transfected as in Fig. 1c. Immunoprecipitated cFLIP_L was analysed by immunoblotting with the indicated antibodies. Protein expression was verified by immunoblotting with the indicated antibodies. **c** HeLa cells were stimulated with or without TNF (10 ng/ml) for 8 h. Cell lysates were immunoprecipitated with anti-cFLIP antibody, and ubiquitylated cFLIP_L was analysed as in Fig. 4b. The upper and lower arrowheads indicate modified and unmodified cFLIP_L, respectively. Asterisks indicate degraded bands of cFLIP_L. **d** WT and *MIB2* KO HeLa cells were immunoprecipitated with control Ig (C) or anti-cFLIP antibody (F), and immunoprecipitated cFLIP_L was analysed as in Fig. 4b. Expression of cFLIP_L and knockout of *MIB2* were analysed by immunoblotting. **e** WT and *MIB2* KO HeLa cells were immunoprecipitated with control or anti-cFLIP antibody. Ubiquitylation of cFLIP_L was analysed by mass spectrometric quantification of ubiquitin chains as described in the Methods. The abundance of each chain was determined by MS analysis. Results are mean ± SD for the pooled results of three independent experiments. Unpaired two-tailed Student *t*-test. ***$P < 0.01$; ND, not detected. **f**, **g** MIB2 ubiquitylates cFLIP_L in vitro. HEK293 cells were transfected with Myc-MIB2, which was immunoprecipitated with anti-Myc antibody. Immunoprecipitates were incubated with ubiquitin, E1, E2 (UbcH5a, UbcH5b, or Ubc13/Uev1a), and GST-cFLIP_L. Ubiquitylated and recombinant cFLIP_L were analysed by immunoblotting with anti-cFLIP (**f**) or K48- and K63-linked polyubiquitin-specific antibodies (**g**). All results are representative of two or three independent experiments.

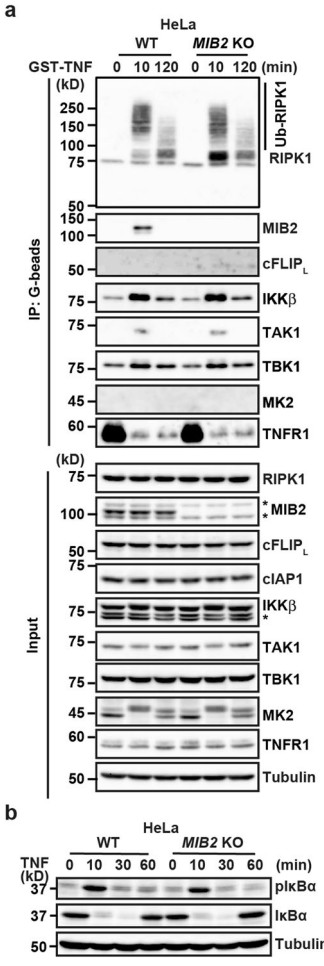

**Fig. 5 TNF-induced complex I formation is not altered in WT and *MIB2* KO cells. a** WT and *MIB2* KO HeLa cells were stimulated with GST-TNF (1 μg/ml) for the indicated times, and TNFR-containing complex I was precipitated with glutathione-Sepharose. Precipitated proteins were analysed by immunoblotting with the indicated antibodies. Protein expression was verified by immunoblotting with the indicated antibodies. **b** WT and *MIB2* KO HeLa cells were stimulated with TNF (10 ng/ml) for the indicated times, and cell lysates analysed by immunoblotting with the indicated antibodies. Asterisks indicate cross-reacted bands. All results are representative of two or three independent experiments.

**MIB2 ubiquitylates multiple sites of cFLIP_L.** To identify which lysine residues of cFLIP_L are ubiquitylated by MIB2, we performed ubiquitylation assays by transiently transfecting HEK293 cells with the deletion mutants of cFLIP_L and HA-Ub, with or without MIB2 (Supplementary Fig. 3a, b). Then, the ubiquitylation of immuno-precipitated cFLIP_L mutants was detected by anti-HA antibody. As expected, MIB2-induced ubiquitylation was abolished in mutants lacking the DM domain, such as cFLIP_L(1–192) or Δ(193–260) (Supplementary Fig. 3a). MIB2 induced ubiquitylation of the N-terminal (1–260) and C-terminal fragments (ΔN193–480) of cFLIP_L, but it did not ubiquitylate GST-cFLIP_L (G193–260) or ΔN193–327 (Supplementary Fig. 3a, b).

Because of the lack of binding, MIB2 did not ubiquitylate cFLIPs, which is identical to the N-terminus of cFLIP_L (Fig. 1d), so we focused on lysine residues in the C-terminus of cFLIP_L. Although mutations of 2–7 lysine residues to arginines at the indicated positions of ΔN193–480 did not abrogate MIB2-dependent K48- or K63-linked polyubiquitylation, mutations of all lysines to arginines (K9R) completely abolished it (Supplementary Fig. 3c). These results indicate that MIB2 attaches K48- or K63-linked polyubiquitin chains to multiple lysine residues in the C-terminus of cFLIP_L. Notably, these 9 lysine residues are highly conserved among various species (Supplementary Fig. 4a); the last four lysines are very adjacent to Q468 in the β6 strand in the C-terminal domain of cFLIP_L that is responsible for dimerisation with caspase 8 (Supplementary Fig. 4b). Thus, ubiquitylation of the last four lysines may affect a proper dimer formation with caspase 8 and subsequent caspase 8 activation.

**TNF-induced complex I formation is not altered in WT and *MIB2* KO cells.** TNF induces the multiprotein complex, termed complex I that mediates NF-κB activation[11,12]. Consistent with a previous study[9], we found that MIB2 was recruited to complex I in WT HeLa cells (Fig. 5a). In sharp contrast, cFLIP_L was not recruited to complex I in either *MIB2* KO or WT HeLa cells. MIB2 bound RIPK1 and cFLIP_L, but cFLIP_L was not recruited to complex I (Fig. 5a), indicating that RIPK1 likely mediates MIB2 recruitment to complex I. Upon TNF stimulation, RIPK1 was highly ubiquitylated in complex I in *MIB2* KO HeLa cells, this polyubiquitylation of RIPK1 was slightly reduced or almost comparable to the level in WT HeLa cells 10 min after stimulation (Fig. 5a). Moreover, IKKβ, TAK1, and TBK1, but not MK2 were recruited to complex I in *MIB2* KO HeLa cells at levels comparable to those in WT HeLa cells after TNF stimulation (Fig. 5a). Consistently, TNF-induced phosphorylation of IκBα and subsequent degradation of total IκBα were not impaired in *MIB2* KO HeLa cells (Fig. 5b), suggesting that TNF-induced NF-κB activation is not impaired in *MIB2* KO cells. Similar results were obtained in *MIB2* KO HCT116 cells (Supplementary Fig. 5a, b, c).

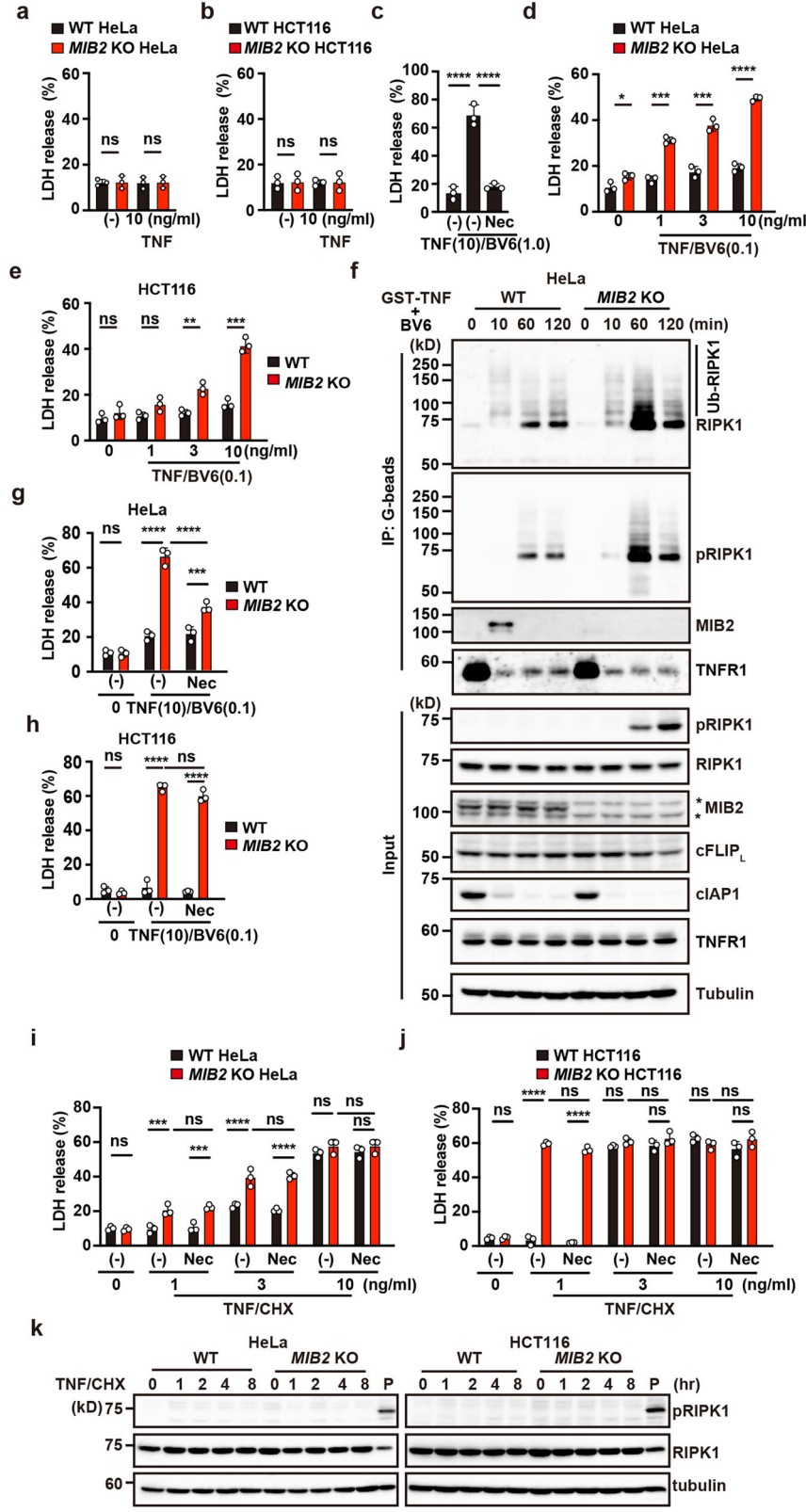

**RIPK1 kinase activity-dependent and -independent apoptosis are enhanced in *MIB2* KO cells**. Next, we investigated whether TNF-induced apoptosis was enhanced in *MIB2* KO HeLa cells. In contrast to *Cflar*-deficient murine embryonic fibroblasts (MEFs) that undergo apoptosis upon TNF treatment alone[19], TNF did not induce apoptosis in WT and *MIB2* KO cells (Fig. 6a, b). In

many types of cells, cIAPs normally prevent TNF-induced cell death through induction of RIPK1 ubiquitylation and recruitment of multiple kinases that negatively regulate RIPK1[25]. Indeed, co-treatment of WT HeLa cells with TNF and an IAP inhibitor, BV6 (1.0 μM) induced apoptosis in WT HeLa cells (Fig. 6c), and this cell death was blocked in the presence of RIPK1 inhibitor,

**Fig. 6 RIPK1 kinase activity-dependent and -independent apoptosis are enhanced in *MIB2* KO cells. a**, **b**, **d**, **e** WT and *MIB2* KO cells were unstimulated, stimulated with TNF (10 ng/ml) (**a**, **b**), or the indicated concentrations of TNF and low concentrations of BV6 (0.1 μM) (**d**, **e**) for 8 h. Cell death was determined by the LDH release assay. Results are mean ± SD of triplicate samples. **c** WT HeLa cells were stimulated with TNF (10 ng/ml)/high concentrations of BV6 (1.0 μM) in the absence or presence of Nec-1s (20 μM) for 8 h. Results are expressed as in **a**. **f** WT and *MIB2* KO HeLa cells were stimulated with GST-TNF (1 μg/ml)/BV6 (0.1 μM) for the indicated times, and the TNFR-containing complex was precipitated and analysed as in Fig. 5a. **g–j** WT and *MIB2* KO cells were unstimulated or stimulated with the indicated concentrations of TNF and low concentrations of BV6 (0.1 μM) (**g**, **h**) or CHX (2.5 μg/ml) (**i**, **j**) in the absence or presence of Nec-1s (20 μM). Results are expressed as in **a**. Unpaired two-tailed Student *t*-test (**a**, **b**, **d**, **e**), one-way ANOVA with Tukey's multiple comparison test (**c**), or two-way ANOVA with Bonferroni's multiple comparison test (**g–j**). *$P < 0.05$; **$P < 0.01$; ***$P < 0.001$; ****$P < 0.0001$; ns, not significant. **k** WT and *MIB2* KO cells were stimulated with TNF (1 ng/ml)/CHX (2.5 μg/ml) for the indicated times. Cell lysates were analysed by immunoblotting with the indicated antibodies. P indicates positive control lysates of *MIB2* KO HeLa cells stimulated with TNF/BV6 (0.1 μM) for 2 h. All results are representative of at least two independent experiments.

Necrostatin-1s (Nec-1s). Intriguingly, although TNF and low concentrations of BV6 (0.1 μM) did not induce apoptosis in WT cells, the treatment did induce apoptosis in *MIB2* KO cells (Fig. 6d, e). Unless otherwise indicated, cells were treated with low concentrations of BV6 (0.1 μM) for subsequent experiments.

A previous study reported that MIB2 suppresses RIPK1 kinase activity and inhibits TNF-induced apoptosis[9], prompting us to test whether TNF/BV6 treatment induced phosphorylation of S166 in RIPK1 in WT and *MIB2* KO HeLa cells. We then stimulated WT and *MIB2* KO HeLa cells with GST-TNF/BV6 and precipitated the TNFR1-containing complex with glutathione beads. As expected, ubiquitylated RIPK1 in the complex was greatly reduced in both types cells 10 min after stimulation, but RIPK1 phosphorylation in the complex was induced 1 h, and then sustained up to 2 h after stimulation (Fig. 6f). RIPK1 phosphorylation in the complex was enhanced in *MIB2* KO HeLa cells. Of note, phosphorylated RIPK1 was detected in the cytosol only in *MIB2* KO HeLa cells, suggesting that RIPK1 phosphorylation in the cytosol may be blocked when RIPK1 binds MIB2 (Fig. 6f). We obtained similar results with *MIB2* KO HCT116 cells (Supplementary Fig. 6).

It was unclear whether MIB2-dependent suppression of apoptosis is mediated solely by suppression of kinase activity of RIPK1 and/or ubiquitylation of cFLIP$_L$. To address this issue, we stimulated *MIB2* KO cells with TNF/BV6 in the absence or presence of Nec-1s. To our surprise, Nec-1s only partially blocked TNF/BV6-induced apoptosis in *MIB2* KO HeLa cells and did not block in *MIB2* KO HCT116 cells (Fig. 6g, h). These results suggest that MIB2 at least in part blocked RIPK1 kinase activity-independent cell death.

To further substantiate our observation, we then treated WT and *MIB2* KO HeLa cells with TNF and the protein synthesis inhibitor cycloheximide (CHX) (TNF/CHX) in the absence or presence of Nec-1s. TNF/CHX treatment mostly induces RIPK1-independent apoptosis[13]; however, a recent study reported that under certain conditions, TNF/CHX also induces RIPK1 kinase activity-dependent apoptosis[15]. In the current work, TNF/CHX-induced cell death was enhanced in *MIB2* KO HeLa cells, especially at low concentrations of TNF (Fig. 6i, j). In addition, these cell death enhancements were not suppressed in the presence of Nec-1s. Consistent with these results, we found that TNF/CHX did not induce phosphorylation of RIPK1 in WT and *MIB2* KO HeLa and HCT116 cells (Fig. 6k). These results further substantiate that MIB2-dependent suppression of TNF-induced apoptosis is not mediated solely by suppression of RIPK1 kinase activity.

Although RIPK1 interacts with MIB1[9], cFLIP$_L$ did not bind MIB1 (Supplementary Fig. 7a, b). Furthermore, knockdown of *MIB1* in WT HeLa cells and *MIB2* KO HeLa cells by siRNA did not further increase TNF-induced cell death (Supplementary Fig. 7c, d).

**MIB2 does not promote degradation of cFLIP$_L$, but rather increases its stability.** cFLIP$_L$ is an unstable protein[19,20], and we

found that MIB2 conjugated K48-linked polyubiquitin chains to cFLIP$_L$, prompting us to test whether MIB2 promotes degradation of cFLIP$_L$. cFLIP$_L$ expression in *MIB2* KO HeLa cells was comparable to the expression in WT cells before and after TNF stimulation alone (Supplementary Fig. 8a). We examined the kinetics of cFLIP expression in the presence of CHX. Although expression levels of cFLIP$_L$ at the early time points were comparable between WT and *MIB2* KO HeLa cells, cFLIP$_L$ completely disappeared by 8 h in *MIB2* KO, but not in WT HeLa cells (Supplementary Fig. 8b). TNF/CHX or TNF/BV6 treatment rapidly induced degradation of cFLIP$_L$ in *MIB2* KO HeLa cells compared to WT HeLa cells (Supplementary Fig. 8c, d). Because TNF/CHX- or TNF/BV6-induced apoptosis was significantly enhanced in *MIB2* KO HeLa cells (Fig. 6d, e, i, j), this rapid degradation might in part be the result of a secondary effect of apoptosis of *MIB2* KO HeLa cells. cFLIP$_S$ was more rapidly degraded after CHX or TNF/CHX treatment than was cFLIP$_L$ (Supplementary Fig. 8b, c), and degradation of cFLIP$_S$ was slightly accelerated in *MIB2* KO HeLa cells compared with WT HeLa cells after CHX stimulation (Supplementary Fig. 8b). Given that MIB2 did not interact with cFLIP$_S$ (Figs. 1d, 2a), the mechanism underlying an increase in cFLIP$_S$ degradation in *MIB2* KO HeLa cells is currently unknown.

In contrast, TNF/BV6 treatment-induced degradation of cFLIP$_S$ in *MIB2* KO, but not WT HeLa cells (Supplementary Fig. 5d). Again, enhanced apoptosis in *MIB2* KO HeLa cells might have contributed to this rapid degradation of cFLIP$_S$. We obtained similar findings in *MIB2* KO HCT116 cells (Supplementary Fig. 8e–h). Together, these results suggest that MIB2 does not promote degradation but rather slightly increases the stability of cFLIP$_L$.

**Complex IIb formation is facilitated in *MIB2* KO HeLa cells.** cFLIP$_L$ is recruited to complex II through homotypic death effector domain (DED) interaction with caspase 8, and promotes or suppresses TNF-induced apoptosis in a context-dependent manner[26,27]. Considering that TNF/BV6 stimulation-induced apoptosis in *MIB2* KO HeLa cells, we tested whether complex IIb formation was facilitated in *MIB2* KO HeLa cells after stimulation. After TNF/BV6 stimulation, anti-FADD antibody immunoprecipitated RIPK1, processed forms of caspase 8, and cFLIP$_L$ in *MIB2* KO HeLa cells, but not WT HeLa cells after TNF/BV6 stimulation (Fig. 7a). Anti-caspase 8 antibody similarly immunoprecipitated RIPK1, FADD, and processed forms of cFLIP$_L$ in *MIB2* KO HeLa cells, but not in WT HeLa cells after TNF/BV6 stimulation (Fig. 7b). Notably, un-ubiquitylated cFLIP$_L$ was recruited to the complex IIb and underwent processing with caspase 8 in *MIB2* KO cell, suggesting that un-ubiquitylated cFLIP$_L$ did not block, but rather promoted caspase 8 activation. Together, MIB2 prevents TNF-induced complex IIb formation. We obtained similar results in *MIB2* KO HCT116 cells (Supplementary Fig. 9a, b).

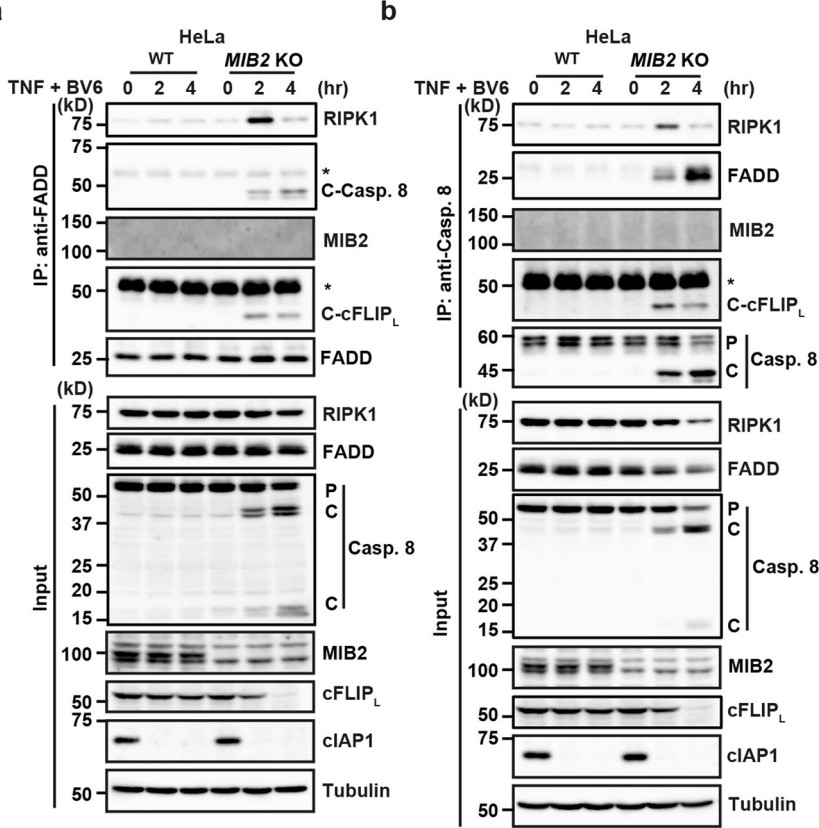

**Fig. 7 Complex IIb formation is facilitated in *MIB2* KO HeLa cells. a, b** WT and *MIB2* KO HeLa cells were stimulated with TNF (10 ng/ml)/BV6 (0.1 μM) for the indicated times, and cell lysates were immunoprecipitated with anti-FADD (**a**) or anti-caspase 8 (**b**) antibodies. Immunoprecipitated proteins were analysed by immunoblotting with the indicated antibodies. Protein expression was verified by immunoblotting with the indicated antibody. *Cross-reacted band; C-Casp. 8, cleaved caspase 8; C-cFLIP$_L$, cleaved cFLIP$_L$. P and C indicate proform and cleaved, respectively. All results are representative of two independent experiments.

**Ubiquitin ligase activity of MIB2 is required for attenuation of TNF-induced apoptosis**. Given that MIB2 binds and then ubiquitylates cFLIP$_L$ (Figs. 1c–f, 2a), whether binding of cFLIP$_L$ to MIB2, ubiquitylation of cFLIP$_L$ per se, or both is required for attenuation of TNF-induced apoptosis is unclear. To discriminate among these possibilities, we confirmed that cFLIP$_L$ still interacted with a mutant MIB2 harbouring mutations in two RING domains (R1R2M; MT for short) and lacking ubiquitin ligase activity (Fig. 8a). We next reconstituted *MIB2* KO HeLa cells with HA-MIB2 WT (*MIB2* KO HeLa/HA-MIB2 WT cells) or HA-MIB2 R1R2M (MT) (*MIB2* KO HeLa/HA-MIB2 MT cells) (Fig. 8b). Consistent with the results in Fig. 3b, ubiquitylation of cFLIP$_L$ was diminished in *MIB2* KO HeLa/HA-MIB2 MT cells compared with *MIB2* KO HeLa/HA-MIB2 WT cells (Fig. 8b). TNF/BV6 or TNF/CHX-induced apoptosis was attenuated in *MIB2* KO HeLa/HA-MIB2 WT cells compared with *MIB2* KO HeLa cells (Fig. 8c–e). In contrast, the TNF/BV6- or TNF/CHX-induced apoptosis in *MIB2* KO HeLa/HA-MIB2 MT cells was comparable to that in *MIB2* KO HeLa cells (Fig. 8c–e). Although Nec-1s only partially suppressed TNF/BV6-induced apoptosis in *MIB2* KO and *MIB2* KO HeLa/HA-MIB2 MT cells, it did not suppress TNF/CHX-induced apoptosis in *MIB2* KO cells and *MIB2* KO HeLa/HA-MIB2 MT cells (Fig. 8d, e). These results substantiate that ubiquitin ligase activity of MIB2 is required for suppression of RIPK1 kinase activity-independent cell death. We obtained similar results in *MIB2* KO HCT116 cells (Fig. 8f–i).

We found that TNF, but not IL-1β, induced dissociation of MIB2 from cFLIP$_L$ after 8 or more hours of stimulation (Supplementary Fig. 10a). However, cFLIP$_L$ released from MIB2 was still ubiquitylated 8 h after TNF stimulation (Fig. 4c). Thus, we assumed that pretreating cells with TNF for at least 8 h and then stimulating them with TNF and BV6 might distinguish whether MIB2 binding to cFLIP$_L$ or ubiquitylation of cFLIP$_L$ per se attenuates TNF-induced apoptosis. TNF/BV6 stimulation did not induce cell death in WT HeLa cells pretreated with TNF for 12 h (Supplementary Fig. 10b). Together, these results further substantiate that ubiquitylation of cFLIP$_L$ per se is required for attenuation of TNF-induced apoptosis, though the mechanism underlying TNF-induced dissociation of cFLIP$_L$ with MIB2 is currently unknown.

**Ubiquitylation of caspase-like domain of cFLIP$_L$ plays a crucial role in attenuating TNF-induced apoptosis**. The results described above cannot formally exclude the possibility that a molecule other than RIPK1 or cFLIP$_L$ mediates the ubiquitylation-dependent anti-apoptotic function of MIB2. To directly demonstrate the contribution of MIB2-mediated cFLIP$_L$ ubiquitylation to the attenuation of TNF-induced apoptosis, we generated a cFLIP$_L$ mutant lacking the ability to bind MIB2. Deletion of the last three amino acids (DCI) in the DM domain of cFLIP$_L$ resulted in abrogated MIB2 binding (Fig. 2f, g), hence, we substituted DCI with alanines, designating this construct as cFLIP$_L$-3A. As expected, cFLIP$_L$-3A lost the ability to interact with MIB2, but the substituted protein still interacted with caspase 8 C/S (Fig. 9a), suggesting that MIB2 may not ubiquitylate cFLIP$_L$-3A. MIB2-dependent ubiquitylation of cFLIP$_L$ was greatly reduced in HEK293 cells transiently transfected with cFLIP$_L$-3A compared with cFLIP$_L$ WT (Fig. 9b). We then reconstituted *Cflar-/-* MEFs

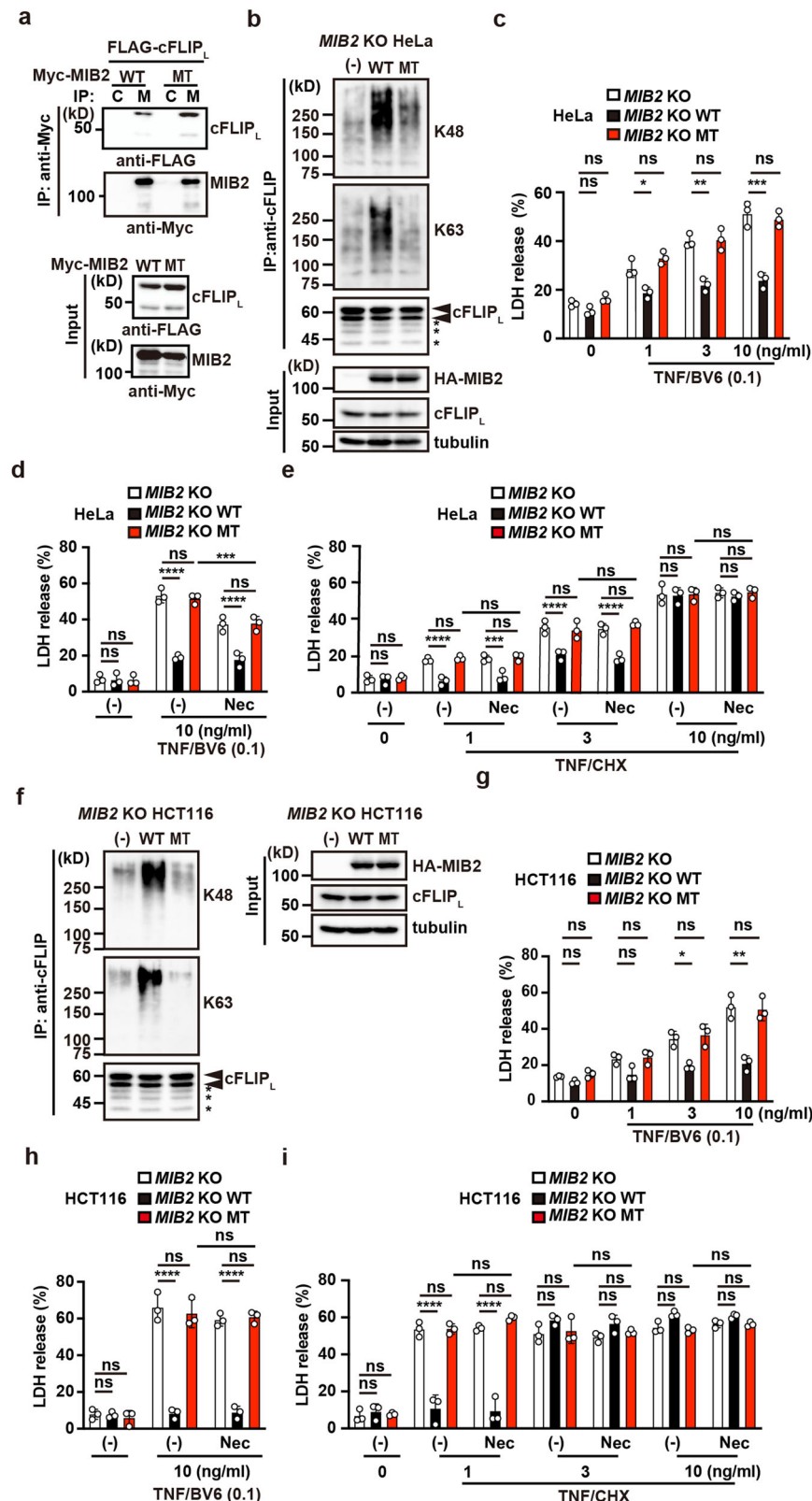

with cFLIP$_L$-3A (*Cflar-/-* cFLIP$_L$-3A cells) or WT cFLIP$_L$ using retroviral vectors. As expected, ubiquitylation of reconstituted cFLIP$_L$-3A was diminished in *Cflar-/-* cFLIP$_L$-3A cells compared with *Cflar-/-* cFLIP$_L$ cells (Fig. 9c). We also found that apoptosis induced by TNF/BV6 was enhanced in *Cflar-/-* cFLIP$_L$-3A cells compared to *Cflar-/-* cFLIP$_L$ cells (Fig. 9d). However,

necroptosis induced by TNF/BV6/zVAD was not enhanced in *Cflar-/-* cFLIP$_L$-3A cells compared with *Cflar-/-* cFLIP$_L$ cells (Fig. 9d).

We next generated a cFLIP$_L$ mutant termed cFLIP$_L$-K9R, in which all nine lysine residues in the caspase-like domain were mutated to arginines (Supplementary Fig. 3c). cFLIP$_L$-K9R

**Fig. 8 Ubiquitin ligase activity of MIB2 is required for attenuation of TNF-induced apoptosis. a** HEK293 cells were co-transfected with FLAG-cFLIP$_L$ and Myc-MIB2 (WT) or Myc-MIB2 R1R2M (MT). Cell lysates were immunoprecipitated with control (C) or anti-Myc (M) antibody, and co-immunoprecipitated proteins were analysed by anti-FLAG antibody. Protein expression was verified by the indicated antibodies using cell lysates. **b, f** *MIB2* KO HeLa (**b**) or *MIB2* KO HCT116 (**f**) cells were reconstituted with HA-MIB2 WT or HA-MIB2 R1R2M (MT). Transfectants were immunoprecipitated with anti-FLIP antibody, and ubiquitylation of cFLIP was determined using anti-K48- or anti-K63-linked polyubiquitin chain-specific antibodies. The upper and lower arrowheads indicate modified and unmodified cFLIP$_L$, respectively. Asterisks indicate degraded bands of cFLIP$_L$. Protein expression was verified by immunoblotting with the indicated antibody. **c–e, g–i** Transfectants of HeLa (**c–e**) or HCT116 (**g–i**) cells expressing the indicated proteins were stimulated with the indicated concentrations of TNF (ng/ml)/BV6 (0.1 μM) (**c, d, g, h**) or TNF/CHX (2.5 μg/ml) (**e, i**) in the absence or presence of Nec-1s (20 μM) for 8 h and cell death was determined by the LDH release assay. Results are mean ± SD of triplicate samples. One-way ANOVA with Tukey's multiple comparison test (**c, g**) or two-way ANOVA with Bonferroni's multiple comparison test (**d, e, h, i**). *$P < 0.05$; **$P < 0.01$; ***$P < 0.001$; ****$P < 0.0001$; ns, not significant. All results are representative of at least two independent experiments.

interacted with MIB2 in a manner comparable to that of cFLIP$_L$-WT (Fig. 9e). We noted that cFLIP$_L$-3A and cFLIP$_L$-K9R also interacted with caspase 8 C/S in a manner comparable to that of WT cFLIP$_L$, indicating that MIB2-mediated ubiquitylation of cFLIP$_L$ did not interfere with cFLIP$_L$ and caspase 8 interaction (Fig. 9a, e). MIB2-dependent ubiquitylation of cFLIP$_L$ was greatly reduced in HEK293 cells transiently transfected with cFLIP$_L$-K9R compared to cFLIP$_L$ WT (Fig. 9f).

We then reconstituted *Cflar-/-* MEFs with cFLIP$_L$-K9R (*Cflar-/-* cFLIP$_L$-K9R cells) or WT cFLIP$_L$ using retroviral vectors. As expected, ubiquitylation of cFLIP$_L$-K9R was diminished in *Cflar-/-* cFLIP$_L$-K9R cells compared with *Cflar-/-* cFLIP$_L$ cells (Fig. 9g). We found that apoptosis induced by TNF/BV6 was enhanced in *Cflar-/-* cFLIP$_L$-K9R cells compared with *Cflar-/-* cFLIP$_L$ cells (Fig. 9h). However, necroptosis induced by TNF/BV6/zVAD was not enhanced in *Cflar-/-* cFLIP$_L$-K9R cells compared with *Cflar-/-* cFLIP$_L$ cells (Fig. 9h). Under these experimental conditions, TNF/BV6 treatment did not induce RIPK1 phosphorylation (Fig. 9i) but did induce cell death (Fig. 9j). Moreover, Nec-1s did not block TNF/BV6-induced cell death in *Cflar-/-* cFLIP$_L$-3A or cFLIP$_L$-K9R cells (Fig. 9j). Together, these results suggest that MIB2-dependent ubiquitylation of cFLIP$_L$ suppresses TNF-induced and RIPK1 kinase activity-independent cell death.

**Ubiquitin ligase activity of MIB2 is dispensable for RIPK1 ubiquitylation, but indispensable for suppression of RIPK1 kinase activity.** To characterise the MIB2-containing complex, we immunoprecipitated this complex from *MIB2* KO HeLa/HA-MIB2 WT cells using anti-HA antibody before and after TNF stimulation. We found that anti-HA antibody precipitated ubiquitylated cFLIP$_L$, RIPK1, and CYLD before TNF stimulation (Fig. 10a). Consistent with the result in Supplementary Fig. 10a, cFLIP$_L$ was released from MIB2, but RIPK1 and CYLD still bound MIB2 8 h after TNF stimulation (Fig. 10a). In contrast, the immunoprecipitates obtained from WT HeLa cells using anti-cFLIP antibody contained MIB2 but not RIPK1 (Fig. 10b), suggesting that the RIPK1/MIB2 complex and cFLIP$_L$/MIB2 complex independently existed in the cells.

To investigate an interplay of ubiquitin ligase activity of MIB2 with RIPK1 ubiquitylation and suppression of RIPK1 kinase activity, we compared ubiquitylation and phosphorylation of RIPK1 in *MIB2* KO HeLa/HA-MIB2 WT and HA-MIB2 MT cells. Ubiquitylation of RIPK1 was not detected before stimulation, but was robustly induced in the MIB2-containing complex precipitated with anti-HA antibody from *MIB2* KO HeLa/HA-MIB2 WT cells or HA-MIB2 MT cells 10 min after TNF stimulation, and declining thereafter (Fig. 10c). These results suggest that the ubiquitin ligase activity of MIB2 does not play a major role in ubiquitylation of RIPK1 after TNF stimulation. In sharp contrast, TNF/BV6 treatment-induced RIPK1 phosphorylation in *MIB2* KO HeLa cells or *MIB2* KO HeLa/HA-MIB2 MT,

but not HA-MIB2 WT cells (Fig. 10d). Together, these results suggest that MIB2's ubiquitin ligase activity plays a crucial role in suppressing RIPK1 kinase activity, but not its ubiquitylation.

When WT cells are stimulated with TNF/IAP inhibitor, autophosphorylation of RIPK1 is prevented by MIB2, therefore RIPK1 cannot undergo complex II formation (Figs. 7, 10e, Supplementary Fig. 9). Although caspase 8 and ubiquitylated cFLIP$_L$ are recruited to the RIPK1/FADD complex, ubiquitylated cFLIP$_L$ prevents a proper oligomer formation with caspase 8, thereby suppressing caspase 8 activation and apoptosis (Fig. 10e). In contrast, when *MIB2* KO cells are stimulated with TNF/IAP inhibitor, autophosphorylation of RIPK1 is induced and phosphorylated RIPK1 subsequently forms the complex IIb (Fig. 7, Supplementary Fig. 9). Recruited un-ubiquitylated cFLIP$_L$ forms a heterodimer with caspase 8 but fails to block caspase 8 activation, thereby inducing apoptosis (Fig. 10e).

**Discussion**

In the present study, we identified MIB2 as an E3 ligase that bound and ubiquitylated cFLIP$_L$. We also found that MIB2 attached K48- and K63-linked polyubiquitin chains to cFLIP$_L$, which was decreased in *MIB2* KO cells. Apoptosis induced by TNF/BV6 and TNF/CHX was enhanced in *MIB2* KO cells, but RIPK1 inhibitor only partially blocked this apoptosis enhancement in *MIB2* KO HeLa cells and did not block it at all in *MIB2* KO HCT116 cells. Recruitment of the caspase 8/cFLIP$_L$ heterodimer to complex IIb and RIPK1 phosphorylation after TNF/BV6 stimulation was facilitated in *MIB2* KO cells, thereby promoting the induction of apoptosis. Moreover, RIPK1 kinase activity-independent apoptosis was induced in cells expressing cFLIP$_L$ mutants lacking MIB2-dependent ubiquitylation. Together, these results indicate that MIB2 suppresses both RIPK1 kinase activity-dependent and -independent apoptosis through suppression of RIPK1 kinase activity and ubiquitylation of cFLIP$_L$, respectively.

cFLIP plays a central role in suppressing death receptor-induced cell death[17,18]. We and other groups have reported that cFLIP$_L$ is rapidly degraded in NF-κB activation-deficient cells[19,20]. Although NF-κB activation upregulates the expression of cFLIP$_L$[28], the ubiquitin/proteasome-dependent pathway degrades cFLIP$_L$. Several E3 ligases, including ITCH, CHIP, and LUBAC, ubiquitylate cFLIP$_L$[20,21,29]. Consistent with a previous study, we found that ITCH ubiquitylates cFLIP$_L$; however, ITCH did not interact with cFLIP$_L$ under our experimental conditions. Hence, ITCH may ubiquitylate cFLIP$_L$ via interaction with another molecule that in turn interacts with cFLIP$_L$. Subsequent analysis identified the minimum amino acids required and sufficient for binding MIB2, referred to as the DM domain. Consistent with the binding experiments, the DM domain is unique to cFLIP$_L$ and not present in cFLIPs or caspase 8. It would be useful to test whether the addition of a cell-permeable peptide containing the DM domain interferes with the interaction

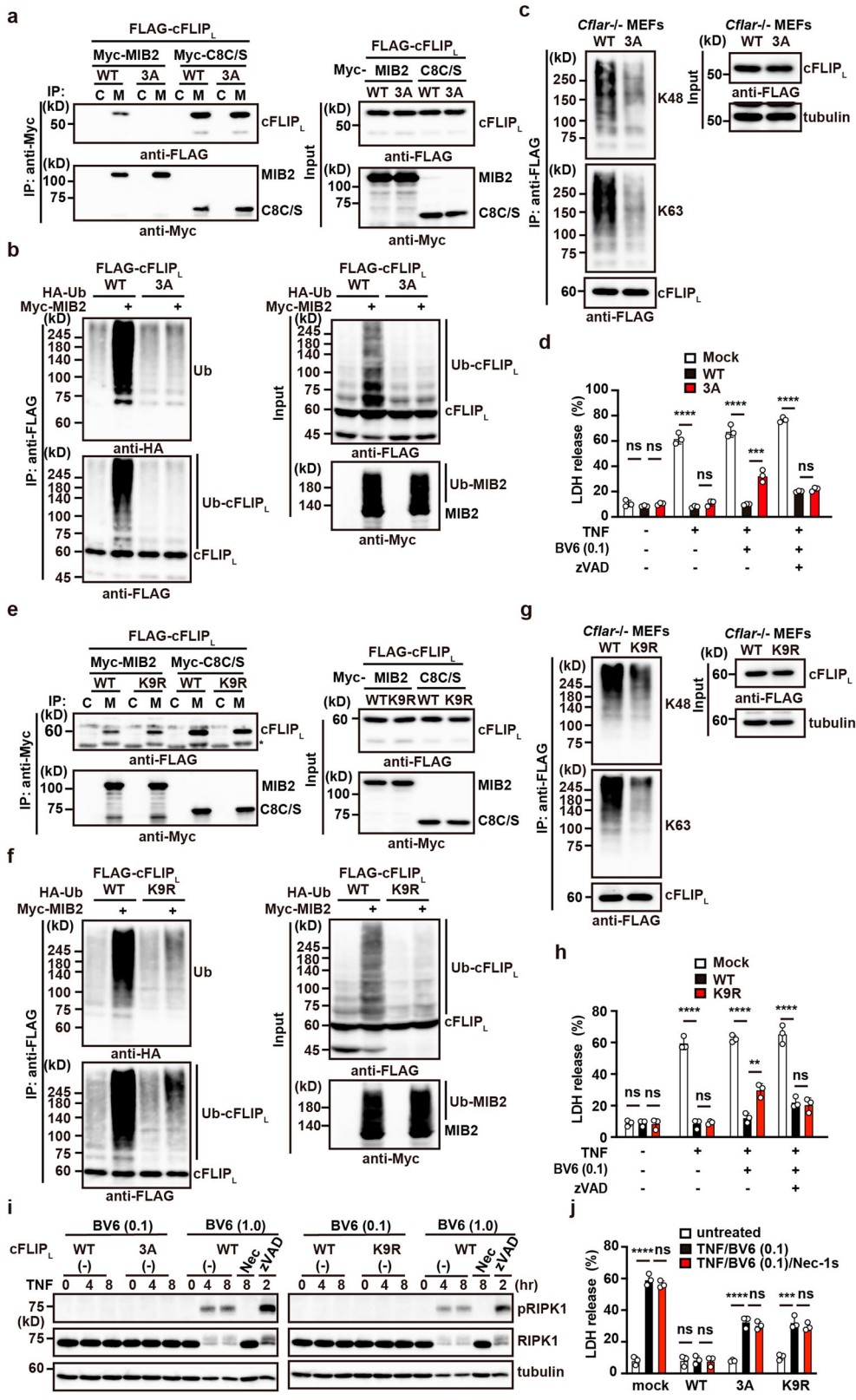

between cFLIP$_L$ and MIB2, thereby increasing susceptibility to TNF-induced apoptosis. Alternatively, it would be interesting to search via chemical library screening for small molecule(s) that interfere with the interaction between MIB2 and cFLIP$_L$ to increase the susceptibility to TNF-induced apoptosis.

MIB2-dependent ubiquitylation of cFLIP$_L$ did not promote and instead blocked degradation of cFLIP$_L$ in the presence of CHX.

Given that K63-linked polyubiquitin chains are involved in signal transduction, but block degradation[23,24], this finding might in part result from MIB2 attachment of K63-linked polyubiquitin chains to cFLIP$_L$. Currently, we cannot formally exclude the possibility that MIB2 independently conjugated K48- and K63-linked polyubiquitin chains to different lysine residues of cFLIP$_L$. Given that the K48–K63 branched chains are refractory to CYLD-

**Fig. 9 Ubiquitylation of the caspase-like domain of cFLIP$_L$ plays a crucial role in attenuating TNF-induced apoptosis. a** cFLIP$_L$-3A interacts with caspase 8 but not MIB2. HEK293 cells were transiently transfected with the indicated expression vectors, and immunoprecipitated proteins were analysed by immunoblotting with the indicated antibodies. WT and 3 A indicate FLAG-cFLIP$_L$ and FLAG-cFLIP$_L$-3A, respectively. C8C/S indicates a protease-inactive mutant of caspase 8. Protein expression was verified by immunoblotting. **b** MIB2-dependent ubiquitylation of cFLIP$_L$-3A is greatly reduced. HEK293 cells were transiently transfected with the indicated expression vectors and analysed as in Fig. 1c. **c** Cflar-/- MEFs were reconstituted with WT or 3 A. Reconstituted Cflar-/- MEFs were immunoprecipitated with anti-FLAG antibody, and ubiquitylation of cFLIP$_L$ was analysed by immunoblotting with anti-K48- or anti-K63-linked polyubiquitin chain-specific antibodies. Expression of reconstituted proteins was verified by immunoblotting. **d** Cflar-/- MEFs reconstituted with an empty vector (mock), WT, or 3 A were stimulated with TNF (10 ng/ml) in the absence or presence of BV6 (0.1 μM) or zVAD (25 μM) for 8 h. Cell viability was determined by LDH release assay. Results are mean ± SD of triplicate samples. **e** cFLIP$_L$-K9R mutant interacts with MIB2 and caspase 8. HEK293 cells were transiently transfected and analysed as in **a**. WT and K9R indicate FLAG-cFLIP$_L$ and FLAG-cFLIP$_L$-K9R, respectively. **f** MIB2-dependent ubiquitylation of cFLIP$_L$-K9R is greatly reduced. HEK293 cells were transiently transfected with the indicated expression vectors and analysed as in Fig. 1c. **g** Reconstituted Cflar-/- MEFs were analysed as in **c**. **h** Reconstituted Cflar-/- MEFs were stimulated and analysed as in **d**. **i** Reconstituted Cflar-/- MEFs were stimulated with TNF (10 ng/ml)/BV6 (0.1 μM) for the indicated times. In parallel experiments, wild-type MEFs were stimulated with TNF (10 ng/ml)/BV6 (1.0 μM) in the absence or presence of Nec-1s (20 μM) or zVAD (25 μM) for the indicated times. Cell lysates were analysed by immunoblotting with the indicated antibodies. **j** Reconstituted Cflar-/- MEFs were stimulated with TNF (10 ng/ml)/BV6 (0.1 μM) in the absence or presence of Nec-1s (20 μM) for the indicated times and analysed as in **d**. One-way ANOVA with Tukey's multiple comparison test (**d**, **h**, **j**). **P < 0.01; ***P < 0.001; ****P < 0.0001; ns, not significant. Results are representative of at least two independent experiments.

mediated deubiquitylation[30], we assume that MIB2 may attach the K48–K63 branched chains to cFLIP$_L$, thereby suppressing deubiquitylation and subsequent K48 chain-mediated degradation. The following experimental data support this hypothesis. First, MIB2-dependent ubiquitylation of cFLIP$_L$ was severely impaired when cells were transfected with K48R or K63R ubiquitin mutants. Second, dissociated cFLIP$_L$ from the MIB2/cFLIP$_L$ complex was still ubiquitylated 8 h after TNF stimulation. This finding suggested that ubiquitylation of cFLIP$_L$ by MIB2 was relatively stable compared to K63-linked polyubiquitin of RIPK1 because ubiquitylation of RIPK1 was rapidly terminated within 2 h after TNF stimulation. On the other hand, our present screening did not identify the E3 ubiquitin ligase that ubiquitylates cFLIP$_L$ and promotes subsequent degradation. We cannot exclude the possibility that ITCH may be responsible for the degradation of cFLIP$_L$. Alternatively, an E3 ligase not included in the present assay, such as HECT, U-box, or PHD finger-type E3 ligases, may be involved in cFLIP$_L$ degradation.

A previous study reported that MIB2 binds and ubiquitylates RIPK1, thereby suppressing heterotypic oligomerization between RIPK1 and FADD via the death domain interaction and TNF-induced apoptosis[9]. We also found that endogenous MIB2 constitutively interacted with RIPK1 and was recruited to complex I. RIPK1 plays a central role in TNF-induced apoptosis[16], but whether the increase in susceptibility to TNF-induced apoptosis was mediated by loss of modification of RIPK1 and/or cFLIP$_L$ in MIB2 KO cells is unclear. The following results strongly support that MIB2-dependent ubiquitylation of cFLIP$_L$ plays a crucial role in the attenuation of TNF-induced apoptosis. First, TNF/BV6-induced cell death was enhanced in MIB2 KO cells, but RIPK1 inhibitor did not block or only partially blocked cell death enhancement in MIB2 KO HCT116 and MIB2 KO HeLa cells, respectively. Second, RIPK1 inhibitor did not block TNF/CHX-induced cell death in MIB2 KO cells. Third, reconstituting Cflar-/- MEFs with cFLIP$_L$ mutants lacking MIB2-dependent ubiquitylation (cFLIP$_L$-3A and cFLIP$_L$-K9R) failed to restore cell viability to levels comparable to cells reconstituted with WT cFLIP$_L$. Fourth, TNF/BV6 treatment of cells expressing cFLIP$_L$-3A or cFLIP$_L$-K9R, but not cFLIP$_L$ WT, resulted in cell death without inducing RIPK1 phosphorylation. Consistent with this result, RIPK1 inhibitor did not block TNF/BV6-induced apoptosis in Cflar-/- cFLIP$_L$-3A or cFLIP$_L$-K9R cells. Together, these findings suggest that ubiquitylation of cFLIP$_L$ contributes, at least in part, to MIB2-dependent suppression of TNF-induced apoptosis in a RIPK1 kinase activity-independent manner.

In contrast to the results of a previous study[9], we found that ubiquitylation of RIPK1 in complex I was not reduced in MIB2 KO HeLa cells or MIB2 KO HCT116 cells after TNF stimulation. Indeed, MIB2 constitutively bound RIPK1 and CYLD before and after TNF stimulation. RIPK1 in the MIB2-containing complex was robustly ubiquitylated in both MIB2 KO HeLa HA-MIB2 WT and HA-MIB2 MT cells 10 min after TNF stimulation and then declined. Moreover, pull-down experiments using glutathione beads after GST-TNF/BV6 stimulation (depleting cIAP1 and cIAP2 proteins) revealed that ubiquitylation of RIPK1 was abolished at early time points, suggesting that ubiquitylation of RIPK1 may be primarily mediated by cIAP1/2. However, we cannot formally exclude the possibility that the difference might result from the relative contribution of MIB1 or MIB2 to ubiquitylation of RIPK1 depending on the cell-type.

In the absence of MIB2, phosphorylation of S$^{166}$, an autophosphorylated serine residue of RIPK1 was enhanced in cells after TNF/BV6 stimulation. Previous studies have shown that IKK, MK2, and TBK1/IKKε phosphorylate RIPK1, attenuating RIPK1 autophosphorylation[15,31–33] and suppressing TNF-induced apoptosis. Consistent with these reports, we found that IKKβ, TAK1, TBK1, but not MK2, were recruited to complex I in WT and MIB2 KO cells after TNF stimulation. Deletion of MIB2 enhanced autophosphorylation of RIPK1 after TNF only in the presence of IAP inhibitor that blocks the activation of IKK and MK2. Hence, MIB2-dependent suppression of RIPK1 kinase activity may be a backup mechanisms for suppressing apoptosis, and it would be intriguing to test whether a kinase other than MK2, IKK, or TBK1/IKKε negatively regulates RIPK1 kinase activity via interaction with MIB2. Pull-down experiments using glutathione beads after GST-TNF/BV6 stimulation revealed that autophosphorylation of RIPK1 increased gradually in both WT and MIB2 KO cells. Of note, phosphorylation of RIPK1 was detected only in the cytosol of MIB2 KO cells and MIB2 KO HA-MIB2 MT cells, but not in WT cells. These results suggest that ubiquitin ligase activity in the MIB2/RIPK1 complex also determines the threshold of expansion of phosphorylated RIPK1 in the cytosol dissociated from the complex containing TNFR1.

The most important finding in the present study is that ubiquitylation of cFLIP$_L$ suppressed TNF-induced apoptosis in a RIPK1 kinase activity-independent manner. Intriguingly, a previous study reported that low concentrations of cFLIP$_L$ promote caspase 8 activation in a cell-free system, whereas high concentrations of cFLIP$_L$ suppress caspase 8 activation[26]. Similar findings were also reported by other groups using cell lines expressing different amounts of cFLIP$_L$ and cFLIPs[34–36]. These results suggest that expression levels of cFLIP$_L$ and cFLIPs might critically determine the threshold for whether cells die or survive depending on the strength of the stimuli of death ligands. In

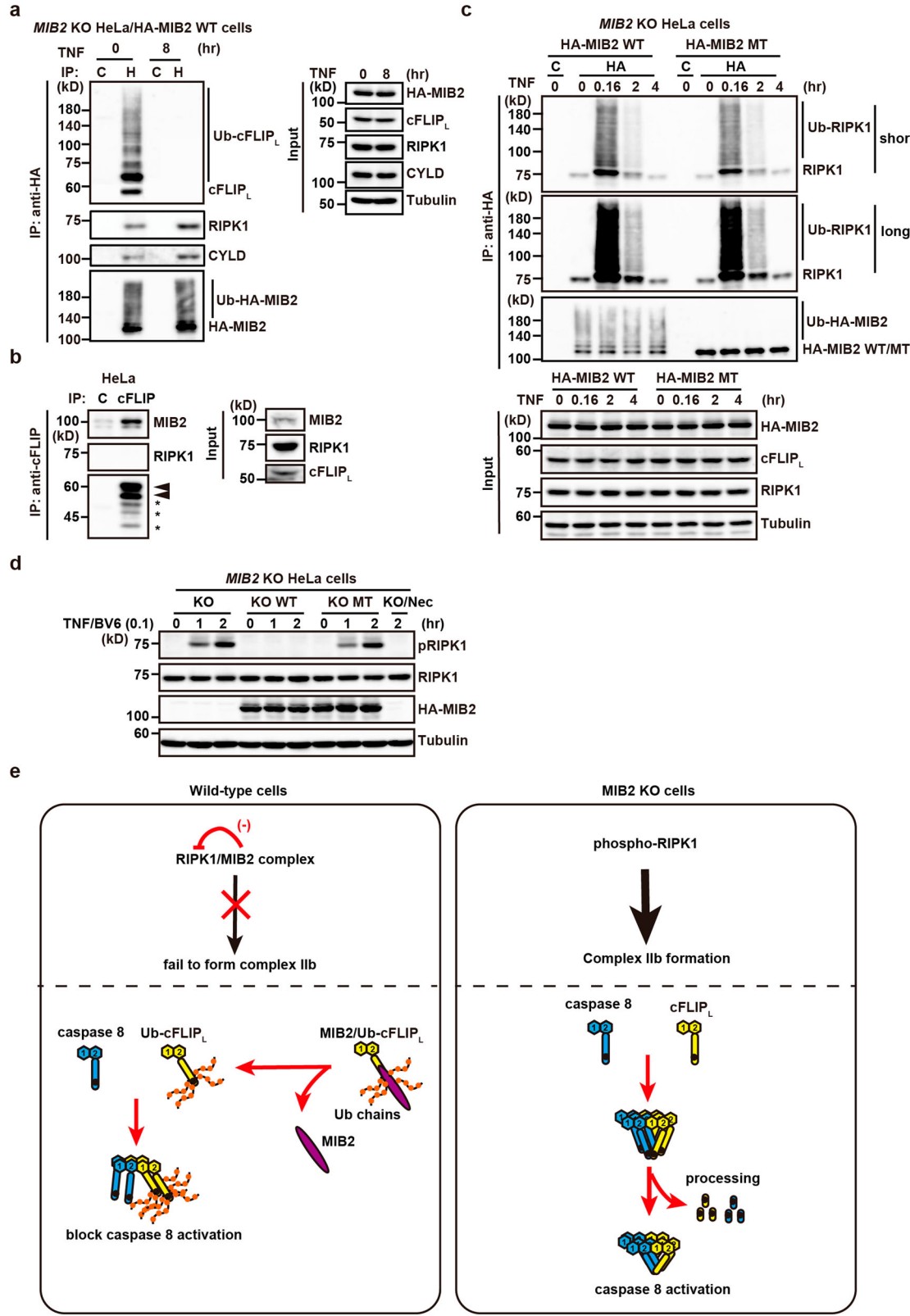

contrast, deletion of *Cflar* results in embryonic lethality because of enhanced cell death[37], indicating that cFLIP is a genuine apoptosis inhibitor in vivo. To integrate these observations obtained from in vitro and in vivo experiments, we need to consider that post-transcriptional modification of $cFLIP_L$ might also affect the anti-apoptotic function of $cFLIP_L$. Indeed, $cFLIP_L$ and cFLIPs are phosphorylated or ubiquitylated[38], although most

studies have focused on the mechanisms underlying their degradation. Previous reports have noted that a Q468D mutant of $cFLIP_L$ fails to form a proper oligomer with caspase 8, preventing caspase 8 processing[26,39]. These results indicate that the C-terminal part of $cFLIP_L$ is required for $cFLIP_L$-dependent caspase 8 activation. In this respect, MIB2-dependent ubiquitylated lysine residues of $cFLIP_L$, including K460, K462, K473, and K474, are

**Fig. 10 Ubiquitin ligase activity of MIB2 is dispensable for RIPK1 ubiquitylation, but indispensable for suppression of RIPK1 kinase activity. a** *MIB2* KO HeLa/HA-MIB2 WT cells were untreated or treated with TNF (10 ng/ml) for the indicated times, and then lysates were immunoprecipitated with control (C) or anti-HA antibody (H). Co-immunoprecipitated proteins were analysed by immunoblotting with the indicated antibodies. Protein expression was verified by the indicated antibodies using cell lysates. **b** The RIPK1/MIB2 complex and cFLIP$_L$/MIB2 complex independently exist in the cells. HeLa cells were immunoprecipitated with control Ig (C) or anti-cFLIP antibody, and co-immunoprecipitated proteins were analysed by immunoblotting with the indicated antibody. The upper and lower arrowheads indicate modified and unmodified cFLIP$_L$, respectively. Asterisks indicate degraded bands of cFLIP$_L$. Protein expression was verified by the indicated antibodies using cell lysates. **c** *MIB2* KO HeLa/HA-MIB2 WT or HA-MIB2 MT cells were untreated or treated with TNF (10 ng/ml) for the indicated times, and lysates were immunoprecipitated and analysed as in **a**. **d** *MIB2* KO (KO), *MIB2* KO HeLa/HA-MIB2 WT (KO WT), or HA-MIB2 MT (KO MT) cells were untreated or treated with TNF (10 ng/ml)/BV6 (0.1 µM) in the absence or presence of Nec-1s (20 µM) for the indicated times. Cell lysates were analysed by immunoblotting with the indicated antibodies. All results are representative of two or three independent experiments. **e** When WT cells are stimulated with TNF/IAP inhibitor, autophosphorylation of RIPK1 is prevented by MIB2, therefore RIPK1 cannot undergo complex IIb formation. Although caspase 8 and ubiquitylated cFLIP$_L$ are recruited to the RIPK1/FADD complex, ubiquitylated cFLIP$_L$ prevents a proper oligomer formation with caspase 8, thereby suppressing caspase 8 activation and apoptosis. In contrast, when *MIB2* KO cells are stimulated with TNF/IAP inhibitor, autophosphorylation of RIPK1 is induced and phosphorylated RIPK1 subsequently forms the complex IIb. Recruited un-ubiquitylated cFLIP$_L$ forms a heterodimer with caspase 8 but fails to block caspase 8 activation, thereby inducing apoptosis.

closely located in Q468. Thus, it is plausible that ubiquitylated cFLIP$_L$ might preferentially interact with and prevent a proper oligomer formation with caspase 8, thereby suppressing caspase 8 activation. Consistent with this idea, cFLIP$_L$ mutants lacking MIB2-dependent ubiquitylation cannot completely restore anti-apoptotic function to the levels found in WT cFLIP$_L$. Indeed, MIB2-dependent suppression of autophosphorylation of RIPK1 plays a crucial role in determining the cell fate through preventing complex II formation. Nonetheless, these data strongly suggest that ubiquitylation of cFLIP$_L$ by MIB2 may critically determine the threshold for whether cells live or die in a RIPK1 kinase activity-independent manner.

## Methods

**Reagents.** Human TNF (BMS301) and murine TNF (34-8321) were purchased from eBioscience; human IL-1β (095-04611) from WAKO; BV6 (CT-BIRI) from Tetralogic Pharmaceuticals; and cycloheximide (C7698), G418 (G5013), polybrene (H9268), and puromycin (P8833) from Sigma-Aldrich; Nec-1s (221984) from abcam. The following antibodies were used in this study: anti-FLAG (M2, Sigma-Aldrich, 1:1000), anti-HA (12CA5, 1:1000; 3F10 for IP, Sigma-Aldrich, 1:1000), anti-Myc (9E10, Sigma-Aldrich, 1:1000), anti-caspase 8 (9746, 1C12, 1:1000; 13423-1-AP, Proteintech, for IP), anti-cFLIP (7F10, Enzo, 1:1000; Dave-2, Adipogen, 1:1000), anti-cIAP1 (1E1-1-10, Enzo, 1:1000), anti-FADD (1F10, MERCK, 1:1000; 14906-1-1-AP, Proteintech, for IP), anti-IκBα (371, C-21, Santa Cruz, 1:1000), anti-phospho-IκBα (2859, 14D4, 1:1000), anti-IKKα/β (7607, Santa Cruz, 1:1000), anti-K48 (Apu2, MERCK, 1:1000), anti-K63 (ab179434, Abcam, 1:1000), anti-MIB1 (D-6, Santa Cruz, 1:1000), anti-MIB2 (ab99378, Abcam, 1:1000), anti-MK2 (3042, 1:1000), anti-phospho-RIPK1 (31122, 1:1000), anti-RIPK1 (610459, BD Transduction Lab, 1:1000), anti-TAK1 (5206, 1:1000), anti-TBK1 (3504, 1:1000), anti-TNFR1 (8436, Santa Cruz, 1:1000), and anti-tubulin (T5168, Sigma-Aldrich, 1:40,000). HRP-conjugated donkey anti-rabbit IgG (NA934, 1:10000), HRP-conjugated goat anti-rat IgG (NA935, 1:10000), and HRP-conjugated sheep anti-mouse IgG (NA931, 1:10000) were obtained from GE Healthcare Life Science. Unless otherwise indicated, antibodies were purchased from Cell Signaling Technology.

**AlphaScreening.** AlphaScreening was performed as described previously[8] and according to the manufacturer's instructions (Perkin Elmer). In the presence of streptavidin-conjugated donor beads and protein A-conjugated acceptor beads together with anti-FLAG antibody, excitation of the donor beads by laser resulted in the generation of singlet oxygen. After an E3 ligase that binds to the donor beads interacts with human cFLIP$_L$, which binds the acceptor beads, and the distance between the two beads is within 200 nm, the singlet oxygen diffuses to react with a chemiluminescent molecule on the acceptor beads, resulting in emission at a wavelength of 520 nm to 620 nm (Fig. 1a). Briefly, 223 E3 ligases containing a RING domain were expressed in wheat cell extracts with biotin-tagged proteins. FLAG-tagged human cFLIP$_L$ and FLAG-tagged dihydrofolate reductors were expressed in the same extracts and used as a bait and a control protein, respectively. Reaction mixtures containing 10 µl of the detection mixture containing 100 mM Tris-HCl (pH 8.0), 100 mM NaCl, 1 mg/ml BSA, 5 µg/ml anti-FLAG antibody, 1 mg/ml BSA, 0.1 µl streptavidin-coated donor beads, and 0.1 µl anti-IgG acceptor beads were added to each well of a 384-well OptiPlate and incubated at 26 °C for 1 h. Luminescence was detected using the AlphaScreen detection programme.

**Cell lines.** HeLa, HCT116, and HEK293 cells were obtained from ATCC. Plat-E cells were provided by T Kitamura. *Cflar-/-* murine embryonic fibroblasts (MEFs; provided by WC Yeh) were described previously[19]. Cells were maintained in DMEM containing 10% fetal calf serum.

**Generation of expression vectors.** All expression vectors used in the present study were constructed using human cDNAs except for ITCH (from mouse cDNA) of the indicated genes. The expression vectors for E3 ligases, including TRIM31, RNF8, LINCR, RNF12, MIB2, and MEX3B, were generated by inserting RT-PCR products into N-terminal 6 Myc-tagged pcDNA3 (provided by K Miyazono). pcDNA3-6Myc-cIAP2 was provided by D Mahoney. The expression vectors for ITCH and caspase 8 C/S were generated by inserting PCR products using pcDNA3-FLAG-ITCH (mouse cDNA; provided by N Tanaka) and pcDNA3-caspase 8 C/S-HA (provided by N Inohara) as templates, respectively, into pcDNA3-6Myc. pCR-FLAG-human cFLIP$_L$, cFLIP$_L$ (1–327), cFLIP$_L$ (1–260), human cFLIPs, and ΔN (193–480) were described previously[19]. ΔN(193–480) mutants containing lysine to arginine mutations at the indicated positions were generated by PCR. GST fusion proteins with N-terminal FLAG tag containing the indicated amino acids from cFLIP$_L$, including G193–260, G193–239, G223–260, G231–260, G239–260, and G239–257, were generated by inserting PCR-amplified fragments into a mammalian expression vector, pCR-FLAG-GST, in which GST-fusion proteins were expressed as N-terminal FLAG tag, using pCR-FLAG-cFLIP$_L$ as a template. A mutant lacking amino acids 139–260 of cFLIP$_L$ (Δ193–260) was generated via recombinant PCR and cloned into pCR-FLAG vector. pCR-FLAG-cFLIP$_L$-3A was generated by recombinant PCR using a standard method. pCR-FLAG-cFLIP$_L$-K9R was generated by replacing the corresponding fragment of wild-type cFLIP$_L$ with a K9R mutant of ΔN(193–480). Deletion mutants of MIB2 containing the indicated amino acids were generated by inserting PCR products into pcDNA3-6Myc vector using pcDNA3-6Myc-MIB2 as a template. pcDNA3-6Myc-MIB2 R1M, R2M, and R1R2M, in which catalytic cysteines were substituted with serines in the first RING (C900S), second RING (C977S), or first and second RING (CC900/977SS) domains, were generated by recombinant PCR. Expression vectors for HA-tagged MIB2 and HA-tagged MIB2 R1R2M mutant were generated by inserting HA-MIB2 and HA-MIB2 R1R2M, respectively, into pCR-HA vector. pCAGGS-HA-Ub and its mutants (K6R, K11R, K27R, K29R, K33R, K48R, and K63R) where lysine at the indicated positions was replaced with arginine, were previously described[40]. Retroviral expression vectors, pMX-Flag-puro that contains a puromycin-resistant gene (provided by T. Kitamura) and pMX-Flag-cFLIP$_L$-puro were described previously[19]. pMX-Flag-cFLIP$_L$-3A-puro and pMX-Flag-cFLIP$_L$-K9R-puro were generated by transferring cFLIP$_L$-3A and cFLIP$_L$-K9R into pMX-Flag-puro, respectively. pEBMulti-MIB2-EGFP, pEBMulti-DsRed-cFLIP$_L$, pEBMulti-DsRed-cFLIPs, and pEBMulti-DsRed-caspase 8 C/S were generated by inserting each PCR-amplified cDNA into pEBMulti vectors (050-0821, Fuji Film). Details of the plasmid constructions will be provided upon request.

**Detection of ubiquitylated proteins.** HEK293 cells ($1.5 \times 10^6$ cells/60-mm dish) were transiently transfected with the indicated expression vectors using PEI MAX 40000 (24765, Polysciences). Twenty-four hours after transfection, cells were lysed with IP buffer (50 mM Tris–HCl [pH 8.0], 150 mM NaCl, 0.5% Nonidet P-40, 25 mM β-glycerophosphate, 1 mM sodium orthovanadate, 1 mM sodium fluoride, 1 mM phenylmethylsulfonyl fluoride [PMSF], 1 mg/ml aprotinin, 1 mg/ml leupeptin, and 1 mg/ml pepstatin). After centrifugation, the supernatant's SDS concentration was adjusted to 1%, and then the supernatants were heated at 95 °C for 10 min to dissociate non-cross-linked protein-protein interactions. After adding IP buffer to dilute the SDS concentration in the supernatant to 0.1% SDS, the supernatants were immunoprecipitated with anti-FLAG antibody overnight at 4 °C. Immunoprecipitates were separated by SDS polyacrylamide gel electrophoresis (SDS-PAGE) and transferred to polyvinylidene difluoride (PVDF) membranes

(IPVH 00010, Millipore). The membranes were immunoblotted with the indicated antibodies and developed with Super Signal West Dura Extended Duration Substrate (34076, Thermo Scientific). The signals were analysed using an Amersham Imager 600 (GE Healthcare Life Sciences).

To detect endogenous ubiquitylated cFLIP$_L$, HeLa, HCT116, Cflar-/-cFLIP$_L$, Cflar-/-cFLIP$_L$-3A, or Cflar-/-cFLIP$_L$-K9R ($4 \times 10^6$ cells/100-mm dish), cells were lysed in IP buffer. After removing insoluble fractions, the subsequent steps were essentially the same as described above. SDS-treated, heat-denatured, and IP buffer-diluted supernatants were immunoprecipitated with anti-cFLIP (for HeLa or HCT116 cells) or anti-FLAG (for Cflar-/- MEFs) antibodies overnight at 4 °C and analysed as described above.

Uncropped images of the results of Western blotting are included in Supplementary Figs. 11–20.

**Co-immunoprecipitation**. HEK293 cells were transiently transfected and lysed with IP buffer as described above, except with further addition of SDS to the supernatants after centrifugation. The supernatants were immunoprecipitated and analysed by immunoblotting with the indicated antibodies. To verify the expression of transfected proteins in cell lysates, the cell lysates were subjected to SDS-PAGE and analysed by immunoblotting as described above.

**Immunofluorescence**. HEK 293 cells were plated on an μ-Dish 35 mm Quad (80416, ibidi) and transiently transfected with expression vectors for MIB2-EGFP, along with DsRed-cFLIP$_L$, DsRed-cFLIPs, or DsRed-caspase8C/S using PEI MAX 40000. The culture media was removed and changed to FluoroBrite™ DMEM (Thermo Fisher Scientific) containing 10% FBS supplemented with antibiotics. Fluorescent images of the cells were collected using a DeltaVision microscope system (GE Healthcare Life Sciences) built on an Olympus IX-71 inverted microscope base equipped with Photometric Coolsnap HQ2 CCD camera. A 100×/NA1.40 UPLS Apo oil immersion lens (Olympus) was used in a 37 °C heat chamber with 5% CO$_2$ gas. FITC and TRITC filter sets were used for collecting data on EGFP and DsRed, respectively. The images were acquired and deconvoluted in SoftWoRx (Applied Precision) and analysed by ImageJ.

**Generation of recombinant proteins**. pGEX-4T-human TNF (hTNF)[41], an expression vector for GST-hTNF (residues 77-233), was provided by ZJ Chen. pGEX-4T-cFLIP$_L$ was generated by inserting full-length cFLIP$_L$ into pGEX-4T vector. GST-hTNF and GST-cFLIP$_L$ were expressed by transforming BL21 with pGEX-4T-hTNF and pGEX-4T-cFLIP$_L$, respectively. The transformants were induced with 0.1 mM IPTG, and the recombinant proteins were purified using glutathione-Sepharose 4B (17075601, GE Healthcare Life Science) and eluted with reduced glutathione (G4251, Sigma-Aldrich) according to the manufacturer's instructions.

**In vitro ubiquitylation**. In vitro ubiquitylation was performed according to the manufacturer's instructions (Boston Biochem). Briefly, cell lysates of HEK293 cells transfected with Myc-MIB2 were immunoprecipitated with anti-Myc antibody. Immunoprecipitated MIB2 derived from one-fourth cell lysates from 100-mm dish was incubated with ubiquitin (10 μM, U-100H), E1 (0.1 μM, E304), E2 (UbcH5a, UbcH5b, or Ubc13/Uev1a) (1 μM, K-980B), and GST-cFLIP$_L$ (0.1 μg) in reaction buffer (50 mM Tris [pH 7.5], 150 mM NaCl, 10 mM MgCl$_2$, 1 mM ATP) at 37°C for 90 min. Reaction mixtures were subjected to SDS-PAGE and analysed by immunoblotting with the indicated antibodies. All reagents except for substrate and E3 were purchased from Boston Biochem.

**Generation of MIB2 KO cells by the CRISPR/Cas9 method**. Two different targeting vectors for human MIB2, termed pX330-MIB2-1 and -2, were generated by inserting two different pairs of oligonucleotides targeting human MIB2 into pX330-U6-Chimeric_BB-CBh-hSpCas9 vector (#42240, Addgene): MIB2-1, 5′- CACCCTG TGCGGTCGGGTGTCGAG-3′ and 5′- AAACCTCGACACCCGACCGCACAG-3′; MIB2-2, 5′- CACCAACTACCGCGCCGGGCTACCA-3′ and 5′- AAACTGG-TAGCCGGCGCGGTAGTT-3′. HeLa and HCT116 cells were transfected with pX330-MIB2-1 or pX330-MIB2-2 using PEI MAX. After limiting dilution, single-cell colonies were expanded and analysed by immunoblotting with anti-MIB2 antibody. We obtained at least two different clones that lost the expression of MIB2 in HeLa and HCT116 cells using two different vectors. We confirmed that MIB2 KO HeLa and HCT116 cells harboured out-of-frame deletions and/or nonsense mutations in MIB2 by genomic sequencing.

**Reconstitution of MIB2 KO HeLa and HCT116 cells**. MIB2 KO HeLa and HCT116 cells were transfected with expression vectors for HA-MIB2 WT or HA-MIB2 R1R2M (MT) using PEI MAX and cultured in 1 mg/ml G418. After 1 week, cells were subject to limiting dilution. Single-cell colonies were expanded and analysed by immunoblotting with anti-HA antibody.

**Mass spectrometric quantification of ubiquitin chains**. Mass spectrometry-based quantification of ubiquitin chains was performed essentially as described previously, but with modifications[30]. WT and MIB2 KO HeLa cells ($1 \times 10^7$ cells/

150-mm dish) were lysed in an IP buffer. After removing insoluble fractions, the subsequent steps were essentially the same as described in "Detection of ubiquitylated protein" section. SDS-treated, heat-denatured, and IP buffer-diluted supernatants were immunoprecipitated with control Ig (for WT HeLa) or anti-cFLIP (for WT or MIB2 KO HeLa cells) antibody overnight at 4 °C. After extensive washing, immunoprecipitates were separated by SDS-PAGE and stained with Bio-Safe Coomassie (Bio-Rad). Gel regions corresponding to molecular weight >50 kDa were excised and extensively washed in 50 mM AMBC/30% acetonitrile (ACN) and 50 mM AMBC/50% ACN. In-gel trypsin digestion was performed by incubation at 37 °C for 15 h with 20 ng/μl Trypsin Gold (Promega) in 50 mM AMBC/5% ACN (pH 8.0). After trypsin digestion, AQUA peptides (12.5 fmol/injection) were added to the extracted peptides, and the concentrated peptides were diluted with 20 μl of 0.1% TFA containing 0.05% H$_2$O$_2$ and incubated at 4 °C overnight. For liquid chromatography-tandem mass spectrometry (LC-MS/MS) analysis, Easy nLC 1200 (Thermo Fisher Scientific) was connected online to Orbitrap Fusion LUMOS (Thermo Fisher Scientific) with a nanoelectrospray ion source (Thermo Fisher Scientific). Peptides were separated using a 45 min gradient (solvent A, 0.1% FA; solvent B, 80% ACN/0.1% FA) with C18 analytical columns (Ionopticks, Aurora Series Emitter Column, AUR2-25075C18A 25 cm × 75 μm, 1.6 μm FSC C18 with nanoZero fitting). For targeted acquisition of MS/MS spectra for ubiquitin chain-derived signature peptides, the Orbitrap Fusion LUMOS instrument was operated in the targeted MS/MS mode by Xcalibur software. The peptides were fragmented by higher-energy collisional dissociation (HCD) with normalised collision energy of 28, and the fragment ions were detected by Orbitrap. Data were processed using PinPoint software 1.3 (Thermo Fisher Scientific), and peptide abundance was calculated based on the integrated area under the curve (AUC) of the selected fragment ions. Mean and standard deviation (SD) were calculated from three biological replicates.

**Degradation of cFLIP$_L$ and cFLIPs in HeLa and HCT116 cells**. Cells were untreated or treated with CHX (2.5 μg/ml), TNF (10 ng/ml), TNF(1 ng/ml)/CHX (2.5 μg/ml), or TNF (10 ng/ml)/BV6 (0.1 μM) for the indicated times, and lysed with a RIPA buffer (50 mM Tris-HCl [pH 8.0], 150 mM NaCl, 1% Nonidet P-40, 0.5% deoxycholate, 0.1% SDS, 25 mM β-glycerophosphate, 1 mM sodium ortho-vanadate, 1 mM sodium fluoride, 1 mM PMSF, 1 μg/ml aprotinin, 1 μg/ml leupeptin, and 1 μg/ml pepstatin). Cell lysates were subjected to SDS-PAGE and analysed by immunoblotting as described above. Signal intensities of cFLIP$_L$, cFLIPs, and tubulin at the indicated times were determined by Image J software and normalised by dividing their intensities by those of tubulin. Relative intensities of normalised intensities of cFLIP$_L$ and cFLIPs at the indicated times compared with those at time 0 (100%) were calculated and are plotted.

**LDH release assay**. Cells were plated onto 96-well plates in DMEM containing 10% FCS. The cells were then stimulated with the indicated concentrations of TNF in the absence or presence of BV6 (0.1 μM), BV6 (1.0 μM), CHX (2.5 μg/ml), Nec-1s (20 μM), zVAD-fmk (25 μM), or respective combinations for 8 h. The LDH release from cells was determined using the Cytotoxicity Detection Kit (Roche) as described previously[42].

**Production of retrovirus**. Retrovirus production and infection were carried out as described previously[43]. Briefly, to produce retrovirus encoding the indicated constructs, we transfected pMX-Flag-puro, pMX-Flag-cFLIP$_L$-puro, pMX-Flag-cFLIP$_L$-3A-puro, or pMX-Flag-cFLIP$_L$-K9R-puro in Plat-E cells using Lipofectamine 2000 (12566014, Invitrogen), and then collected the culture supernatant. Cflar-/- MEFs were infected with the culture supernatant in the presence of polybrene (10 μg/ml) and then incubated in puromycin (2.5 μg/ml) to isolate stable transfectants. After confirming the expression of the transfected gene, pooled puromycin-selected fibroblasts were used for further experiments.

**Induction of dissociation of cFLIP$_L$ with MIB2, and TNF-induced cytotoxicity assay**. HeLa and HCT116 cells ($4 \times 10^6$ cells/100-mm dish) were treated with TNF (10 ng/ml) or IL-1β (10 ng/ml) for the indicated times. Cells were lysed with IP buffer, and cell lysates were immunoprecipitated with anti-cFLIP antibody. Immunoprecipitated proteins were analysed by immunoblotting with anti-MIB2 or anti-cFLIP antibodies. WT and MIB2 KO HeLa cells were untreated or pretreated with TNF (10 ng/ml) for 12 h, and then cells were stimulated with the indicated concentrations of TNF (ng/ml)/BV6 (0.1 μM) for 8 h. Cell death was determined by LDH release assay.

**Isolation of complex I and complex IIb**. To isolate complex I, cells ($4 \times 10^6$/100-mm dish) were stimulated with GST-TNF (1 μg/ml) or GST-TNF (1 μg/ml)/BV6 (0.1 μM) for the indicated times. Cells were lysed with IP buffer. After centrifugation, cell lysates were precipitated with glutathione-Sepharose overnight at 4 °C. After washing with IP buffer, the precipitates were subjected to SDS-PAGE and analysed by immunoblotting with the indicated antibodies. To isolate complex IIb, cells ($4 \times 10^6$/100-mm dish) were stimulated with TNF (10 ng/ml)/BV6 (0.1 μM) for the indicated times. After centrifugation, cell lysates were immuno-precipitated with anti-FADD or caspase 8 antibodies overnight at 4 °C. After washing, the precipitates were subjected to SDS-PAGE and analysed by

immunoblotting with the indicated antibodies. Expression of the indicated proteins (input) was analysed by immunoblotting with the indicated antibodies.

**Knockdown of cFLIP or MIB1 by siRNAs.** WT HeLa cells were transfected with non-target (D-001810-10-05 from GE Healthcare Life Science) or *CFLAR* siRNA (5′-GGAGCAGGGACAAGUUACA-3′ from Qiagen) using Lipofectamine 2000 (11668027, Thermo Fisher Scientific)[44]. Sixteen hours after transfection, cells were immunoprecipitated with anti-cFLIP antibody (7F10), and immunoprecipitates were analysed by anti-cFLIP antibody (Dave-2). The knockdown efficiency of cFLIP was determined by immunoblotting with anti-cFLIP antibody using total cell lysates. To knockdown MIB1, WT, and *MIB2* KO HeLa cells were transfected with non-target (D-001810-10-05) or *MIB1* siRNAs (L-014033-00-0005 from GE Healthcare Life Science) as described above. Sixteen hours after transfection, cells were plated onto 96-well plates and stimulated with TNF (10 ng/ml) and BV6 (0.1 μM) for 8 h. Cell death was determined by LDH release assay. The knockdown efficiency of MIB1 was determined by immunoblotting cell lysates with anti-MIB1 antibody.

**Statistical and reproducibility.** Statistical analyses were performed by unpaired two-tailed Student's *t*-test, one-way ANOVA with Tukey's multiple comparisons test, or two-way ANOVA with Bonferroni's multiple comparisons test. $P < 0.05$ was considered significant.

AlphaScreening was performed in duplicates. LDH release assay was performed in triplicates. Quantification of different linkage of ubiquitin chains by liquid chromatography-tandem mass spectrometry analysis was pooled results of three independent experiments. All results, including LDH release assay, confocal microscopy analysis, and Western blotting are representative of two or three independent experiments.

**Reporting Summary.** Further information on research design is available in the Nature Research Reporting Summary linked to this article.

## Data availability

The authors declare that the data supporting the findings of this study are available within the paper and its supplementary information files. Source data behind the graphs are available as Supplementary Data 2–8. Data not included are available from the corresponding authors upon reasonable request.

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

## Acknowledgements

We thank WC Yeh for the *Cflar-/-* MEFs, ZJ Chen for pGEX-4T-hTNF, K Miyazono for the pcDNA3-6Myc vector, N Tanaka for pcDNA3-FLAG-ITCH, N Inohara for pcDNA3-caspase 8 C/S-HA, and T Kitamura for Plat-E cells and the pMX-puro vector. This work was supported in part by Grants-in-Aid for Scientific Research (B) 17H04069 (to H.N.) and (B) 20H03475 (to H.N.), Scientific Research (C) 20K09231 (to O.N.), and Challenging Exploratory Research 17K19533 (to H.N.) from Japan Society for the Promotion of Science (JSPS), Scientific Research on Innovative Areas 26110003 (to H.N.), the Japan Agency for Medical Research and Development through AMED-CREST (Grant no. 20gm1210002; to H.N.), and a Private University Research Branding project (to H.N.) from the Ministry of Education, Culture, Sports, Science, and Technology, Japan, and Toho University Grant for Research Initiative Program (TUGRIP) from Toho University, Japan.

## Author contributions

O.N., K.M., and H.N. designed the study and interpreted the results. O.N. performed and analysed most experiments. H.T. and T.S. designed and performed AlphaScreening. S.K-S. and Y.K. performed biochemical experiments. S.M. performed fluorescence imaging analysis. F.O., Y.S., and Y.Y. performed Mass spectrometric quantification of ubiquitin chains. O.N., S.K-S., and Y.T. constructed plasmids. S.Y. and F.T. supervised the experiments. F.T. provided critical reagents. O.N., K.M., and H.N. wrote the manuscript with constructive input from all authors.

## Competing interests

The authors declare no competing interests.
