## [Peer Review File · Communications Biology]

Reviewers' comments:

Reviewer #1 (Remarks to the Author):

This is a very important study, which identifies a new E3 ligase, MIB2, that ubiquitinylates cFLIPL and shows that it specifically binds to this protein at the C-terminal part as well as it ubiquitinylates this protein at the C-terminal and very N-terminal part. Importantly, they compare the effects of this kinase with ITCH and interestingly find that ITCH seems to have very minor effects on the ubiquitinylation of cFLIPL. They very elegantly show that this is a specific kinase for cFLIPL because it does not interact with cFLIPS. Moreover, they show that MIB2 binds to the caspase-like part of cFLIPL and ubiquitinylates the very C-terminal part of cFLIPL. Further, the authors analyse the role of this ubiquitinylation event in TNF-induced cell death. The study is supported by a number of very sophisticated biochemical experiments. Despite all these very important points, I have several major comments on the manuscript and on the mechanism suggested.

Major comments:

1. Ubiquitinylation of cFLIPL seems to be essential for the prevention of its interaction with procaspase-8 and formation of the DED filaments. However, the authors identify the very C-terminal part of cFLIPL as a main target of ubiquitinylation. However, this part should play a secondary role in the recruitment of c-FLIPL to the DED filament, as the DED part of cFLIPL plays an essential role here. This hypothesis does not fit to the current knowledge on the DED filament formation

2. Why procaspase-8 can not form DED filaments without cFLIP and bind to FADD? There are many studies that show that this is possible, i.e. Hughes et al, 2016, Mol Cell and others

3. Then if we assume that cFLIPL is not recruited to the DED filaments, there is also an essential role which belongs to c-FLIPS in controlling the activity of the filament, cFLIPS will be still present in the system. The role of both isoforms have to be considered. Finally, cFLIPL plays both anti- and pro-apoptotic role in the DED filament, which is not accounted in this model.

4. C-FLIP seem to undergo K48 and K63 ubiquitinylation by MIB2 (Highlighted in Figure 4). Moreover, the authors applied a very elegant quantitative mass spectrometry approach in order to quantify the degree of the ubiquitinylation of cFLIPL. However, it has to be said that surprisingly, the absence of MIB2 decreased K48 and K63 ubiquitinylation of cFLIPL only two fold. Hence, my question is how the two times difference in the degree of ubiquitilation would have such dramatic influence on the functional outcome?

5. Along these lines, in HeLa cells cFLIP is expressed only to the lower levels. Hence, I do not see how this low level expression of cFLIPL and the two-times decrease of its K63 and K48 levels can lead to such a drastic differences in apoptosis and formation of complex II upon KO of MIB2

6. Figure 7: Apparently TNF/BV6 treatment does not cause the formation of complex II in HeLa cells, which contradicts several reports. More detailed titration and kinetics have to be done, as it is rather not likely. Many other groups reported the assembly of the complex II in the similar conditions and the cells are dying which means they have to have complex II

7. Figures 6,7, the levels of cFLIPL are not changed plus minus MIB2, this indeed rules out any

regulation of apoptosis via expression levels of cFLIP and probably leaves the room only for checking out the role for K63 chains.

Summarizing, there is no evidence (experimental or theoretical) presented that it is indeed ubiquitination of c-FLIP leads to the different sensitivity of HeLa MIB2 KO cells I would rather assign the observed effects to the effects of MIB2 on RIPK1. Hence the functional role of c-FLIP ubiquitination by MIB2 still have to be uncovered

Minor comments:

8. Extended figure 3: in the quantification of c-FLIP levels the loading control is overexposed, hence, it is difficult to draw the conclusion. C-FLIPs is always quicker degraded, which was also observed in both cell lines (KO Mib2 versus wt) so one can not draw a conclusion that this is yet another evidence of MIB2 action

9. In TNF signaling complex I is not termed 'death inducing signaling complex' but just complex I. With complex IIa and IIb, there are certainly some different views on this nomenclature in the literature, as some people name complex IIb as necrosome or RIPK1/RIPK3 complex and complex IIa as ripoptosome, but in a particular review that was cited by authors indeed complex IIb is assigned to FADD/caspase-8/c-FLIP complex and complex IIa to TRADD/caspase-8/FADD complex. I personally prefer the first nomenclature.

Reviewer #2 (Remarks to the Author):

Nakabayashi report c-FLIP-long as a novel binding partner and substrate of the E3 ligase MIB-2. MIB-2-dependent ubiquitylation of c-FLIP-long is shown to suppress TNF-induced apoptosis. The conclusions are largely justified by the data, although the notion that ubiquitylation is affecting c-FLIP stability is less certain. For example, unmodified c-FLIP may just be more prone to aggregation than degradation in extended fig. 3. Do they get the same result if they examine the insoluble fraction for c-FLIP?

My other main criticisms are:

- 1) It is not clear anywhere in the text, methods, or legends if experiments exclusively use human rather than mouse proteins.
- 2) Throughout the figures, the efficiency of each IP is not confirmed by WB for the protein that is immunoprecipitated. These quality control WBs should be included in every experiment.
- 3) The authors should indicate exactly which cysteines are mutated in R1M and R2M MIB-2.
- 4) Fig. 3C should include WBs using linkage-specific Ub antibodies (K48, K63, and K11 antibodies). The linkages generated in this more defined in vitro setting would be useful to compare to those generated in cells.
- 5) The c-FLIP input WB in Fig. 5f should be less cropped so that the reader can see the loss of ubiquitylation on the 3A mutant (as compared to the WT control). i.e. are the slower migrating forms absent?

Reviewer #3 (Remarks to the Author):

In this manuscript, Nakabayashi et al. show that E3 ligase Mind bomb 2 (MIB2) can bind and generate K48/K63-linked ubiquitin chains on cFLIPL. Previous study has shown MIB2 could ubiquitinate RIPK1 to inhibit RIPK1 kinase activity, thereby suppress TNF α -induced apoptosis (Feltham et al., 2018). However, the authors found cells expressing MIB2-binding defective mutant of cFLIPL are also hypersensitive to TNF α -induced apoptosis. Therefore, they draw the conclusion that MIB2 inhibits TNF-induced cell death through both ubiquitination of cFLIPL and inhibition of RIPK1 kinase activity. Overall, this study identified cFLIPL as a new substrate of E3 ligase MIB2 based on in vitro biochemistry data, and extended the current understanding of the function of MIB2 in the regulation of TNF α -induced apoptosis. However, more experiments need to be performed to determine the molecular mechanism of how MIB2-mediated ubiquitination of cFLIPL suppresses TNF α -induced apoptosis independent of RIPK1 kinase activity inhibition.

Specific comments:

1. Line 65, TAK1 inhibitors are thought to inhibit the phosphorylation but not ubiquitination of RIPK1.

2. In Fig. 1c, d, total HA-Ub and IP-Flag level were missing.

3. Extended Fig. 2c, the data is not sufficient to demonstrate that MIB2 ubiquitinates multiple lysine residues of cFLIPL. Since MIB2 could generate both K63- and K48-linked ubiquitin chains on cFLIPL, it may not be easy to detect the change of total ubiquitination level. The authors should also detect the K63- and K48-linked ubiquitination level in various mutants.

4. Extended Fig. 3a, b, the MIB2 on the stability of cFLIPL seems to have no much significance. This experiment is important and needs to be improved. Since the authors show MIB2 could generate K63- and K48-linked ubiquitination on cFLIPL, it is critical to make sure the role of MIB2 on the stability of cFLIPL, since MIB2 has been shown to be involved in TNF α -induced apoptosis through regulating RIPK1. Therefore, the author should examine the stability of cFLIPL in WT and MIB2 KO cells under different apoptosis inducers (TNF alone; CHX alone; TNF+CHX; TNF+BV-6). In addition, cFLIPL is degraded very faster under apoptosis condition. The dose and time point of stimulators should be modified.

5. Fig. 5a-5e, deficiency of MIB2 causes TNF α -induced apoptosis due to enhanced RIPK1 kinase activity has been reported earlier (Feltham et al., 2018). It is critical to discriminate the relationship between MIB2-mediated ubiquitination of cFLIPL and MIB2-mediated suppression of RIPK1 kinase activity. Therefore, the authors should use RIPK1 kinase inhibitor or genetically knockout RIPK1 in MIB2-KO cells to determine the contribution of ubiquitination of cFLIPL to the apoptosis in MIB2-KO cells. In addition, the authors need to examine the RIPK1 kinase activity in cFLIPL-3A-reconstituted MEFs to determine whether MIB2-mediated ubiquitination of cFLIPL could suppress RIPK1 kinase activation.

6. Fig. 6c, 6f, although reconstitution of MIB2 KO cells with MIB2-MT (Ligase activity defective) was comparable to the apoptosis in MIB2 KO cells, it is not convincing to conclude defective ubiquitination of cFLIPL mediated by MIB2 is required for TNF α -induced apoptosis in MIB2 KO cells, since MIB2 could also ubiquitinate RIPK1. More experiments are needed to determine whether RIPK1 kinase-dependent cell death is dominant in MIB2-MT reconstitution cells. First, the authors

should examine whether MIB2-MT can still interact and ubiquitinate RIPK1. Second, the cell death assay should add a control using RIPK1 kinase inhibitor (eg, Necrostatin-1) to examine the contribution of RIPK1 kinase activity in the cell death of MIB2-MT reconstitution cells.

7. Fig. 6h, the cFLIPL level in the immunoprecipitates was missing; and an IgG control should be added to exclude the non-specific binding.

8. Extended Fig. 5a, it is weird that IP-C, cFLIPL blotting has IgGs, but in IP-H, cFLIPL blotting nearly has no IgGs.

9. Line 367, a spelling error “compalex”

10. Line 342-343, although TNF plus CHX is initially thought to induce RIPK1-independent cell death. However, under some conditions such as TBK1 deficiency, it could also induce RIPK1 kinase-dependent cell death (Xu et al., 2018). Since MIB2 could suppress RIPK1 kinase activation, TNF plus CHX may also induce RIPK1 kinase-dependent cell death in MIB2 KO cells. The authors should examine RIPK1 autophosphorylation and use RIPK1 kinase inhibitor to do the TNF plus CHX stimulation in MIB2 KO cells.

11. Line 378-382, K63-linked ubiquitination of RIPK1 has been reported to suppress RIPK1 kinase activity by recruiting TAK1 kinase (Tang et al., 2019). The authors should examine the TAK1, IKK, TBK1 and MK2 recruitment in TNFR1 complex1 in MIB2-KO cells to determine the exact molecular mechanism by which MIB2-mediated ubiquitination suppresses RIPK1 kinase activity.

In this manuscript, Nakabayashi et al. show that E3 ligase Mind bomb 2 (MIB2) can bind and generate K48/K63-linked ubiquitin chains on cFLIP_L. Previous study has shown MIB2 could ubiquitinate RIPK1 to inhibit RIPK1 kinase activity, thereby suppress TNF α -induced apoptosis (Feltham et al., 2018). However, the authors found cells expressing MIB2-binding defective mutant of cFLIP_L are also hypersensitive to TNF α -induced apoptosis. Therefore, they draw the conclusion that MIB2 inhibits TNF-induced cell death through both ubiquitination of cFLIP_L and inhibition of RIPK1 kinase activity. Overall, this study identified cFLIP_L as a new substrate of E3 ligase MIB2 based on in vitro biochemistry data, and extended the current understanding of the function of MIB2 in the regulation of TNF α -induced apoptosis. However, more experiments need to be performed to determine the molecular mechanism of how MIB2-mediated ubiquitination of cFLIP_L suppresses TNF α -induced apoptosis independent of RIPK1 kinase activity inhibition.

Specific comments:

1. Line 65, TAK1 inhibitors are thought to inhibit the phosphorylation but not ubiquitination of RIPK1.
2. In Fig. 1c, d, total HA-Ub and IP-Flag level were missing.
3. Extended Fig. 2c, the data is not sufficient to demonstrate that MIB2 ubiquitinates multiple lysine residues of cFLIP_L. Since MIB2 could generate both K63- and K48-linked ubiquitin chains on cFLIP_L, it may not be easy to detect the change of total ubiquitination level. The authors should also detect the K63- and K48-linked ubiquitination level in various mutants.
4. Extended Fig. 3a, b, the MIB2 on the stability of cFLIP_L seems to have no much significance. This experiment is important and needs to be improved. Since the authors show MIB2 could generate K63- and K48-linked ubiquitination on cFLIP_L, it is critical to make sure the role of MIB2 on the stability of cFLIP_L, since MIB2 has been shown to be involved in TNF α -induced apoptosis through regulating RIPK1. Therefore, the author should examine the stability of cFLIP_L in WT and MIB2 KO cells under different apoptosis inducers (TNF alone; CHX alone; TNF+CHX; TNF+BV-6). In addition, cFLIP_L is degraded very faster under apoptosis condition. The dose and time point of stimulators should be modified.
5. Fig. 5a-5e, deficiency of MIB2 causes TNF α -induced apoptosis due to enhanced RIPK1 kinase activity has been reported earlier (Feltham et al., 2018). It is critical to discriminate the relationship between MIB2-mediated ubiquitination of cFLIP_L and MIB2-mediated suppression of RIPK1 kinase activity. Therefore, the authors should use RIPK1 kinase inhibitor or genetically knockout RIPK1 in MIB2-KO cells to determine the contribution of ubiquitination of cFLIP_L to the apoptosis in MIB2-KO cells. In addition, the authors need to examine the RIPK1 kinase activity in

cFLIP_L-3A-reconstituted MEFs to determine whether MIB2-mediated ubiquitination of cFLIP_L could suppress RIPK1 kinase activation.

6. Fig. 6c, 6f, although reconstitution of MIB2 KO cells with MIB2-MT (Ligase activity defective) was comparable to the apoptosis in MIB2 KO cells, it is not convincing to conclude defective ubiquitination of cFLIP_L mediated by MIB2 is required for TNF α -induced apoptosis in MIB2 KO cells, since MIB2 could also ubiquitinate RIPK1. More experiments are needed to determine whether RIPK1 kinase-dependent cell death is dominant in MIB2-MT reconstitution cells. First, the authors should examine whether MIB2-MT can still interact and ubiquitinate RIPK1. Second, the cell death assay should add a control using RIPK1 kinase inhibitor (eg, Necrostatin-1) to examine the contribution of RIPK1 kinase activity in the cell death of MIB2-MT reconstitution cells.

7. Fig. 6h, the cFLIP_L level in the immunoprecipitates was missing; and an IgG control should be added to exclude the non-specific binding.

8. Extended Fig. 5a, it is weird that IP-C, cFLIP_L blotting has IgGs, but in IP-H, cFLIP_L blotting nearly has no IgGs.

9. Line 367, a spelling error “compalex”

10. Line 342-343, although TNF plus CHX is initially thought to induce RIPK1-independent cell death. However, under some conditions such as TBK1 deficiency, it could also induce RIPK1 kinase-dependent cell death (Xu et al., 2018). Since MIB2 could suppress RIPK1 kinase activation, TNF plus CHX may also induce RIPK1 kinase-dependent cell death in MIB2 KO cells. The authors should examine RIPK1 autophosphorylation and use RIPK1 kinase inhibitor to do the TNF plus CHX stimulation in MIB2 KO cells.

11. Line 378-382, K63-linked ubiquitination of RIPK1 has been reported to suppress RIPK1 kinase activity by recruiting TAK1 kinase (Tang et al., 2019). The authors should examine the TAK1, IKK, TBK1 and MK2 recruitment in TNFR1 complex1 in MIB2-KO cells to determine the exact molecular mechanism by which MIB2-mediated ubiquitination suppresses RIPK1 kinase activity.

Reviewer #1 (Remarks to the Author):

This is a very important study, which identifies a new E3 ligase, MIB2, that ubiquitinylates cFLIPL and shows that it specifically binds to this protein at the C-terminal part as well as it ubiquitinylates this protein at the C-terminal and very N-terminal part. Importantly, they compare the effects of this kinase with ITCH and interestingly find that ITCH seems to have very minor effects on the ubiquitinylation of cFLIPL. They very elegantly show that this is a specific ligase for cFLIPL because it does not interact with cFLIPS. Moreover, they show that MIB2 binds to the caspase-like part of cFLIPL and ubiquitinylates the very C-terminal part of cFLIPL. Further, the authors analyse the role of this ubiquitinylation event in TNF-induced cell death. The study is supported by a number of very sophisticated biochemical experiments. Despite all these very important points, I have several major comments on the manuscript and on the mechanism suggested.

RESPONSE: Thank you very much for your appreciation of our study and critical comments to improve our manuscript.

Major comments:

1. Ubiquitinylation of cFLIPL seems to be essential for the prevention of its interaction with procaspase-8 and formation of the DED filaments. However, the authors identify the very C-terminal part of cFLIPL as a main target of ubiquitinylation. However, this part should play a secondary role in the recruitment of c-FLIPL to the DED filament, as the DED part of cFLIPL plays an essential role here. This hypothesis does not fit to the current knowledge on the DED filament formation

RESPONSE: Thank you for pointing out a critical issue. Indeed, N-terminal DED domains of cFLIP_L are required for DED filament formation with caspase 8. However, C-terminal domain of cFLIP_L also mediates the interaction with caspase 8 (Shu et al., 1997; Yu et al., 2009). Moreover, as reported in previous studies (Hughes et al., 2016; Yu et al., 2009), substitution of glutamine at 468 with glutamic acid (Q468D) in the C-terminal caspase-like domain of cFLIP_L fails to promote the processing of recombinant caspase 8 in a cell-free system.

These results suggest that the C-terminal part of cFLIP_L is involved in a proper heterodimer formation with caspase 8, and subsequent caspase 8 activation. Notably, multiple lysine residues in the C-terminal part of cFLIP_L, including K460, K462, K473, and K474, are adjacent to Q468 (Supplementary Fig. 4a, b); thus, ubiquitylation of these lysine residues by MIB2 might block complex IIb maturation through steric hindrance. Consistent with the hypothesis, cFLIP_L mutants lacking MIB2-dependent ubiquitylation, such as cFLIP_L-3A or cFLIP_L-K9R cannot completely restore anti-apoptotic function to levels found in WT cFLIP_L (Fig. 9d, h). These results further substantiate a crucial role for C-terminal caspase-like domain of cFLIP_L in attenuating caspase 8 activation. We made a new Fig. 10 to represent our model and mentioned the results in the text (line 206-211, 341-378, 402-411, 465-478, 518-527).

2. Why procaspase-8 can not form DED filaments without cFLIP and bind to FADD? There are many studies that show that this is possible, i.e. Hughes et al, 2016, Mol Cell and others.

RESPONSE: Indeed, in the absence of cFLIP, caspase 8 undergoes homodimer and forms DED filaments, resulting in induction of apoptosis (Dillon et al., 2012; Panayotova-Dimitrova et al., 2013; Piao et al., 2012; Yeh et al., 2000). However, as reported by others (Hughes et al., 2016; Yu et al., 2009), in the presence of cFLIP_L, caspase 8 preferentially interacts with cFLIP_L, but not caspase 8 itself. This might be because the closed L2' loop conformation of caspase 8 causes steric clash when the proform of caspase 8 undergoes homodimer (Yu et al., 2009). Thus, ubiquitylated cFLIP_L might preferentially interact with caspase 8, but block higher-ordered oligomer formation and caspase 8 activation. Given that immunoprecipitates with anti-FADD and anti-caspase 8 antibodies did not contain caspase 8 and FADD, respectively, in WT HeLa cells treated with TNF/BV6 (0.1 μM) (see Fig. 7a, b, and Supplementary Fig. 9a, b), FADD and caspase 8 did not appear to form a stable complex in the presence of ubiquitylated cFLIP_L. We made a new Fig. 10 to represent our model and mentioned it in the text (line 402-411, 507-527).

3. Then if we assume that cFLIPL is not recruited to the DED filaments, there is also an essential role which belongs to c-FLIPS in controlling the activity of the filament, cFLIPS will be still present in the system. The role

of both isoforms have to be considered. Finally, cFLIPL plays both anti- and pro-apoptotic role in the DED filament, which is not accounted in this model.

RESPONSE: Thank you for pointing out a critical issue. cFLIP_L and cFLIPs are expressed in human cells such as HeLa and HCT116 cells. Overexpression of cFLIPs and viral homologs of cFLIPs (v-FLIPs) blocks death receptor-induced apoptosis in murine and human cells (Thome et al., 1997). However, expression levels of cFLIPs were comparable to those of cFLIP_L in HeLa and HCT116 cells (Supplementary Fig. 8). Moreover, cFLIPs was also recruited to complex IIb along with cFLIP_L (Rebuttal Figure 1)(Hillert et al., 2020). It is unclear whether cFLIPs is recruited to complex IIb in the absence of cFLIP_L. Although overexpression of cFLIPs blocks apoptosis, but increases susceptibility to necroptosis in vitro and in vivo (Oberst et al., 2011; Shindo et al., 2019), it is unclear whether endogenous cFLIPs in the absence of cFLIP_L plays a major role in cell fate decision. Relative contribution of cFLIP_L and cFLIPs to TNF-induced apoptosis may be evaluated by generating cells specifically lacking cFLIP_L or cFLIPs by *CRISPR/Cas9* technology. However, such experiments are beyond the scope of the present study and will be addressed in the future study. We only discussed these points in the rebuttal letter.

NakabayashiRebuttalFig1

Rebuttal Fig. 1 Both cFLIP_L and cFLIPs are recruited to TNF-induced complex II. WT and *MIB2* KO HeLa cells were stimulated with TNF + BV6 (0.1 mM) for the indicated times. TNF-induced complex II was immunoprecipitated

with anti-caspase 8 antibody and detected with anti-cFLIP antibody. TL indicates total cell lysates derived from untreated WT HeLa cells. The upper and lower arrowheads indicate cFLIP_L and cFLIPs, respectively. Asterisk indicates cross-reacted heavy chain bands of immunoprecipitated antibodies.

Regarding the function of cFLIP_L, Hughes et al. elegantly showed a model that low concentrations of cFLIP_L promote activation of caspase 8 through promoting processing of caspase 8, whereas high concentrations of cFLIP_L suppress caspase 8 activation (Hughes et al., 2016). This conclusion is drawn by the experiments using in vitro translated cFLIP_L in a cell-free system. In sharp contrast, germline deletion of *Cflar* results in embryonic lethality due to enhanced apoptosis and necroptosis (Yeh et al., 2000), indicating that cFLIP is a genuine apoptosis inhibitor in vivo.

To integrate these apparently inconsistent observations obtained from in vitro and in vivo experiments, we surmised that post-transcriptional modification of cFLIP_L that occurs in cells but not in a cell-free system might critically affect the function of cFLIP_L. Q468D mutant of cFLIP_L fails to form a proper oligomer with caspase 8, preventing caspase 8 processing (Hughes et al., 2016; Yu et al., 2009). These results indicate that the C-terminal part of cFLIP_L is required for cFLIP_L-dependent caspase 8 activation. In this respect, MIB2-dependent ubiquitylated lysine residues of cFLIP_L, including K460, K462, K473, and K474, are closely located in Q468. Thus, it is plausible that Ubiquitylated cFLIP_L and caspase 8 heterodimers fail to form higher-ordered oligomers through steric hindrance, thereby blocking complex IIb maturation and caspase 8 activation. Consistent with this idea, cFLIP_L mutants lacking MIB2-dependent ubiquitylation cannot completely restore anti-apoptotic function to the levels found in WT cFLIP_L. Thus, these results suggest that in addition to relative amounts of cFLIP_L vs. caspase 8 described above, posttranscriptional modification of cFLIP_L may critically determine whether cell live or die. Thus, our study has provided a new avenue leading to complete understanding of regulation of anti-apoptotic and proapoptotic functions of cFLIP_L. We mentioned these points in the text and made new Figs. 9 and 10 to include these results (line 341-378, 402-411, 507-527).

4.C-FLIP seem to undergo K48 and K63 ubiquitylation by MIB2

(Highlighted in Figure 4). Moreover, the authors applied a very elegant quantitative mass spectrometry approach in order to quantify the degree of the ubiquitinylation of cFLIPL. However, it has to be said that surprisingly, the absence of MIB2 decreased K48 and K63 ubiquitinylation of cFLIPL only two fold. Hence, my question is how the two times difference in the degree of ubiquitylation would have such dramatic influence on the functional outcome?

RESPONSE: Thank you for pointing out a critical issue. Although total amounts of K48's and K63's ubiquitylation of cFLIP_L was only reduced to half, it is plausible that a decrease in ubiquitylation of critical lysine residues to prevent a higher-ordered oligomer formation and caspase 8 activation (Fig. 10e, f). Indeed, reconstitution of *Cflar*^{-/-} MEFs with cFLIP_L-3A or cFLIP_L-K9R (both mutants lack MIB2-dependent ubiquitylation) failed to suppress TNF/BV6-induced apoptosis comparable to the levels in cFLIP_L WT cells (Fig. 9). Moreover, there may be some threshold of ubiquitylated cFLIP_L that can suppress complex IIb formation in response to TNF along with concentrations of BV6. Indeed, TNF and low concentrations of BV6 (0.1 μM) induced apoptosis in *MIB2* KO cells, but not WT cells, whereas TNF and high concentrations of BV6 (1.0 μM) induced apoptosis in both WT and *MIB2* KO cells (Fig. 6c-j). These results indicate that the MIB2-dependent ubiquitylation of cFLIP_L may critically determine the threshold for whether cells live or die (line 230-240, 341-378, 507-527).

5. Along these lines, in HeLa cells cFLIP is expressed only to the lower levels. Hence, I do not see how this low level expression of cFLIPL and the two-times decrease of its K63 and K48 levels can lead to such a drastic differences in apoptosis and formation of complex II upon KO of MIB2.

RESPONSE: Although the expression levels of cFLIP are very low, knockdown, or knockout of cFLIP greatly enhances TNF-induced cell death (Nakajima et al EMBO J 2006; Nakajima et al Oncogene 2008). Please also see the response to the comment 4.

6. Figure 7: Apparently TNF/BV6 treatment does not cause the formation of

complex II in HeLa cells, which contradicts several reports. More detailed titration and kinetics have to be done, as it is rather not likely. Many other groups reported the assembly of the complex II in the similar conditions and the cells are dying which means they have to have complex II.

RESPONSE: Thank you for picking up this. Concentrations of BV6 may critically determine whether TNF/BV6 induces complex IIb formation and apoptosis in various cell types. Notably, low concentrations of BV6 (0.1 μM) with TNF did induce complex IIb formation and apoptosis in *MIB2* KO cells, but not wild-type cells (Fig. 6, 7, and Supplementary Fig. 9). In sharp contrast, high concentrations of BV6 (1 μM) with TNF induced complex IIb formation and cell death in wild-type cells (Fig. 6c and Rebuttal Fig. 2). Although low concentrations of BV6 (0.1 μM) was sufficient to completely deplete cIAP1/2 (Fig. 6f, 7a, b, Supplementary Fig. 6), high concentrations of BV6 (1 μM) were required to induce the complex IIb formation and apoptosis in wild-type cells (Fig. 6c and Rebuttal Fig. 2). Elucidation of the mechanisms underlying BV6 concentration-dependent induction of complex IIb is a very important issue but is beyond the scope of the present study. We mentioned it in the text (line 228-257).

NakabayashiRebuttalFig2

Rebuttal Fig. 2 High, but not low concentrations of BV6 along with TNF induces complex II formation. WT HeLa cells were stimulated with TNF (10 ng/ml) in the absence or presence of the indicated concentrations of BV6. TNF-induced complex IIb was immunoprecipitated with anti-caspase 8 antibody and detected by anti-RIPK1 antibody. Asterisk indicates cross-reacted bands. Results are representative of two independent experiments.

7. Figures 6,7, the levels of cFLIP_L are not changed plus minus MIB2, this indeed rules out any regulation of apoptosis via expression levels of cFLIP and probably leaves the room only for checking out the role for K63 chains.

RESPONSE: Thank you for your thoughtful comments. Although MIB2 ubiquitylated cFLIP_L, MIB2-dependent ubiquitylation did not promote degradation of cFLIP_L, but slightly increased the stability of cFLIP_L (Supplementary Fig. 8). As suggested, the most important finding in the present study is that MIB2-dependent ubiquitylation of cFLIP_L blocks complex IIb maturation, thereby suppressing caspase 8 activation. cFLIP_L mutants lacking MIB2-dependent ubiquitylation, such as cFLIP_L-3A and cFLIP_L-K9R cannot completely restore anti-apoptotic function to levels found in WT cFLIP_L after TNF/BV6 stimulation (Fig. 9d, h). Moreover, despite the lack of RIPK1 activation, TNF/BV6 induced apoptosis in *Cflar*^{-/-} cells expressing cFLIP_L-3A or cFLIP_L-K9R, but not cFLIP_L WT (Fig. 9i). Consistent with these results, RIPK1 inhibitor did not block TNF/BV6-induced apoptosis in *Cflar*^{-/-} cFLIP_L-3A or cFLIP_L-K9R cells (Fig. 9j). Collectively, these results strongly support that ubiquitylation of cFLIP_L suppresses RIPK1 kinase activity-independent cell death. We made a new Fig. 9 to include these results and mentioned these points in the text (line 341-378, 507-527).

Summarizing, there is no evidence(experimental or theoretical) presented that it is indeed ubiquitylation of c-FLIP leads to the different sensitivity of HeLa MIB2 KO cells I would rather assign the observed effects to the effects of MIB2 on RIPK1. Hence the functional role of c-FLIP ubiquitylation by MIB2 still have to be uncovered

RESPONSE:

Indeed, as described above, we also found that MIB2 suppressed RIPK1-dependent apoptosis in a ubiquitin ligase-activity dependent manner. In contrast to a previous study, TNF-induced RIPK1 ubiquitylation was not impaired in *MIB2* KO cells. In addition, our present study strongly supports the idea that MIB2 suppresses RIPK1 kinase activity-independent cell death through ubiquitylation of cFLIP_L. The following results strongly support the idea. First, TNF/BV6-induced cell death was enhanced in MIB2 KO cells, but RIPK1

inhibitor did not block or only partially blocked cell death enhancement in MIB2 KO HCT116 and MIB2 KO HeLa cells, respectively. Second, RIPK1 inhibitor did not block TNF/CHX-induced cell death in MIB2 KO cells. Third, reconstituting *Cflar*^{-/-} MEFs with cFLIP_L mutants lacking MIB2-dependent ubiquitylation (cFLIPL-3A and cFLIPL-K9R) failed to restore cell viability to levels comparable for cells reconstituted with WT cFLIP_L. Fourth, TNF/BV6 treatment of cells expressing cFLIP_L-3A or cFLIP_L-K9R, but not cFLIP_L WT, resulted in cell death without inducing RIPK1 phosphorylation. Consistent with this result, RIPK1 inhibitor did not block TNF/BV6-induced apoptosis in *Cflar*^{-/-} cFLIP_L-3A or cFLIP_L-K9R cells. Together, these findings suggest that ubiquitylation of cFLIP_L might mediate MIB2-dependent suppression of TNF-induced apoptosis in a RIPK1 kinase activity-independent manner. We made new Figures 6, 8, 9, 10, and mentioned these points in the text (line 460-478)

Minor comments:

8. Extended figure 3: in the quantification of c-FLIP levels the loading control is overexposed, hence, it is difficult to draw the conclusion. C-FLIPs is always quicker degraded, which was also observed in both cell lines (KO Mib2 versus wt) so one can not draw a conclusion that this is yet another evidence of MIB2 action.

RESPONSE: As suggested, we performed additional experiments to investigate degradation of cFLIP_L after stimulation with different stimuli in more detail. Signal intensities of cFLIP_L, cFLIPs, and tubulin were determined by Image J, and ratios of cFLIP_L/tubulin and cFLIPs/tubulin were calculated. Relative intensities of each ratio at the indicated times compared to the ratio at 0 (100 %) was calculated. Degradation of cFLIP_L in *MIB2* KO cells was slightly accelerated but not delayed compared to WT cells after CHX or TNF/CHX stimulation (Supplementary Fig. 8). Rapid degradation of cFLIP_L and cFLIPs in *MIB2* KO cells after TNF/BV6 stimulation might in part be the result of a secondary effect of apoptosis of *MIB2* KO cells. We mentioned these in the text and included the results to make a new Supplementary Fig. 8 (line 273-295).

9. In TNF signaling complex I is not termed 'death inducing signaling complex' but just complex I. With complex IIa and IIb, there are certainly some different views on this nomenclature in the literature, as some

people name complex IIb as necrosome or RIPK1/RIPK3 complex and complex IIa as ripoptosome, but in a particular review that was cited by authors indeed complex IIb is assigned to FADD/caspase-8/c-FLIP complex and complex IIa to TRADD/caspase-8/FADD complex. I personally prefer the first nomenclature.

RESPONSE: Indeed, there is some discrepancy of nomenclatures of TNFR1 complex IIa, IIb, necrosome, and ripoptosome. To avoid confusion, we mentioned that “the death-inducing signaling complex, which is referred to as complex IIb or ripoptosome” in the text (line 70).

Reviewer #2 (Remarks to the Author):

Nakabayashi report c-FLIP-long as a novel binding partner and substrate of the E3 ligase MIB-2. MIB-2-dependent ubiquitylation of c-FLIP-long is shown to suppress TNF-induced apoptosis. The conclusions are largely justified by the data, although the notion that ubiquitylation is affecting c-FLIP stability is less certain. For example, unmodified c-FLIP may just be more prone to aggregation than degradation in extended Fig. 3. Do they get the same result if they examine the insoluble fraction for c-FLIP?

RESPONSE: We appreciate positive comments on our manuscript and pointing out a critical issue. As suggested, we prepared insoluble fractions by adding sample buffer to the pellets of cell lysates after lysis with RIPA buffer followed by sonication. Then, solubilized insoluble fractions were subjected to SDS-PAGE. As shown in Rebuttal Fig. 3, we could not detect cFLIP_L in insoluble fractions. Thus, it is unlikely that unmodified cFLIP_L forms aggregates, thereby disappearing from cell lysates after treatment. We only mentioned it in the rebuttal letter.

NakabayashiRebuttalFig3

Rebuttal Figure 3 Un-ubiquitylated cFLIP_L does not translocate into insoluble fractions after stimulation. WT and MIB2 KO cells were stimulated

with TNF/CHX for the indicated times. Soluble and insoluble fractions were prepared and analyzed by immunoblotting with the indicated antibodies. P indicates positive control samples of cell lysates of untreated WT HeLa and HCT116 cells.

Regarding the stability of cFLIP_L, we assume that slightly increased stability of cFLIP_L does not play a major role in preventing TNF-induced cell death (Supplementary Fig. 8). But rather, ubiquitylation of cFLIP_L in the C-terminal caspase-like domain suppresses caspase 8 activation through binding and preventing higher-ordered oligomer formation and caspase 8 activation. We mentioned it in the text and present a model in Fig. 10e, f (line 402-411, 507-527).

My other main criticisms are:

1) It is not clear anywhere in the text, methods, or legends if experiments exclusively use human rather than mouse proteins.

RESPONSE: Sorry for the lack of description. We used all human cDNAs except for ITCH (from mouse cDNA) for constructing expression vectors in the experiments. We mentioned it in the Methods (line 569-570).

2) Throughout the figures, the efficiency of each IP is not confirmed by WB for the protein that is immunoprecipitated. These quality control WBs should be included in every experiment.

RESPONSE: Sorry for not including critical control experiments. As suggested, we included the results showing the efficiency of immunoprecipitated proteins with the indicated antibodies for IP experiments.

3) The authors should indicate exactly which cysteines are mutated in R1M and R2M MIB-2.

RESPONSE: Sorry for the lack of a detailed description of MIB2 mutants. We mutated catalytic cysteine(s) at 900 and 977 to serine(s) in the RING finger domains of MIB2. We mentioned these in the text, Methods, and Figure legends (line 147-148, 589-592, 942-943) .

4) Fig. 3C should include WBs using linkage-specific Ub antibodies (K48, K63, and K11 antibodies). The linkages generated in this more defined in vitro setting would be useful to compare to those generated in cells.

RESPONSE: As suggested, we performed Western blotting with anti-K48- or anti-K63-linked polyubiquitin chain-specific antibodies using products from in vitro ubiquitylation assays. Indeed, anti-K11-linked polyubiquitin chain-specific antibody had been commercially available from Merck (MABS107) until recently. However, the vendor has stopped supplying the antibody. Thus, we only included the results with anti-K48- or anti-K63-linked polyubiquitin chain-specific antibodies in Fig. 4g and mentioned it in the text (line 183-189).

5) The c-FLIP input WB in Fig. 5f should be less cropped so that the reader can see the loss of ubiquitylation on the 3A mutant (as compared to the WT control). i.e. are the slower migrating forms absent?

RESPONSE: Thank you for picking up this. Since we did not co-transfect HA-Ub along with WT or cFLIP_L-3A and MIB2 in old Fig. 5f, we did not detect a slower migrating form of cFLIP_L in the uncropped blot (Rebuttal Fig. 4). To show a defect in MIB2-dependent ubiquitylation of cFLIP_L-3A, we transfected WT and cFLIP_L-3A along with MIB2 and HA-Ub. As shown in Fig. 9b, MIB2-dependent ubiquitylation of cFLIP_L-3A was greatly reduced in HEK293 cells compared to the levels in WT cFLIP_L. We mentioned it in the text (line 351-353).

NakabayashiRebuttal Fig4

Reviewer #3 (Remarks to the Author):

In this manuscript, Nakabayashi et al. show that E3 ligase Mind bomb 2 (MIB2) can bind and generate K48/K63-linked ubiquitin chains on cFLIPL. Previous study has shown MIB2 could ubiquitinate RIPK1 to inhibit RIPK1 kinase activity, thereby suppress TNF α -induced apoptosis (Feltham et al., 2018). However, the authors found cells expressing MIB2-binding defective mutant of cFLIPL are also hypersensitive to TNF α -induced apoptosis. Therefore, they draw the conclusion that MIB2 inhibits TNF-induced cell death through both ubiquitination of cFLIPL and inhibition of RIPK1 kinase activity. Overall, this study identified cFLIPL as a new substrate of E3 ligase MIB2 based on in vitro biochemistry data, and extended the current understanding of the function of MIB2 in the regulation of TNF α -induced apoptosis. However, more experiments need to be performed to determine the molecular mechanism of how MIB2-mediated ubiquitination of cFLIPL suppresses TNF α -induced apoptosis independent of RIPK1 kinase activity inhibition.

Thank you for your positive comments on our study. As suggested, we performed substantial amounts of experiments to address the comments raised by the reviewers. Now the results presented here strongly support that MIB2-dependent ubiquitylation of cFLIPL suppresses RIPK1 kinase activity-independent cell death.

Specific comments:

1. Line 65, TAK1 inhibitors are thought to inhibit the phosphorylation but not ubiquitination of RIPK1.

RESPONSE: We appreciate pointing out our misunderstanding. As suggested, we changed the sentence to "IAP inhibitors and TAK1 inhibitors that block ubiquitylation and phosphorylation of RIPK1, respectively" (line 67-68).

2. In Fig. 1c, d, total HA-Ub and IP-Flag level were missing.

RESPONSE: As suggested, we included the results in Fig. 1c and 1d.

3. Extended Fig. 2c, the data is not sufficient to demonstrate that MIB2 ubiquitinates multiple lysine residues of cFLIPL. Since MIB2 could generate both K63- and K48-linked ubiquitin chains on cFLIPL, it may not be easy to detect the change of total ubiquitination level. The authors should also detect the K63- and K48-linked ubiquitination level in various mutants.

RESPONSE: Thank you for your valuable suggestion. As suggested, we performed Western blotting with anti-K48- or anti-K63-linked polyubiquitin chain-specific antibodies using various mutants. Combined mutations of the indicated lysine residues in the C-terminal caspase-like domain did not change the signals reacted with anti-K48- or anti-K63-linked polyubiquitin chain-specific antibodies, suggesting that both K48-linked and K63-linked polyubiquitin chains were anchored to each lysine (Supplementary Fig. 3c). Moreover, we also performed Western blotting with anti-K48- or anti-K63-linked polyubiquitin chain-specific antibodies using products of in vitro ubiquitylation assays (Fig. 4g). We mentioned these results in the text (line 183-189, 202-211).

4. Extended Fig. 3a, b, the MIB2 on the stability of cFLIPL seems to have no much significance. This experiment is important and needs to be improved. Since the authors show MIB2 could generate K63- and K48-linked ubiquitination on cFLIPL, it is critical to make sure the role of MIB2 on the stability of cFLIPL, since MIB2 is involved in TNF α -induced apoptosis through regulating RIPK1. Therefore, the author should examine the stability of cFLIPL in WT and MIB2 KO cells under different apoptosis inducers (TNF alone; CHX alone; TNF+CHX; TNF+BV-6). In addition, cFLIPL is degraded very faster under apoptosis condition. The dose and time point of stimulators should be modified.

RESPONSE: As suggested, we performed additional experiments to investigate kinetics of degradation of cFLIP_L and cFLIPs after stimulation with different stimuli in more detail. Signal intensities of cFLIP_L, cFLIPs, and tubulin were determined by Image J, and cFLIP_L/tubulin and cFLIPs/tubulin ratios were calculated. Relative percentages of each ratio at the indicated times compared

to the ratio at 0 (100%) were calculated. Degradation of cFLIP_L in *MIB2* KO cells was slightly accelerated but not delayed compared to those in WT cells after CHX or TNF/CHX stimulation (Supplementary Fig. 8). This rapid degradation might in part be the result of a secondary effect of apoptosis of *MIB2* KO HeLa cells. We mentioned these results in the text and made a new Supplementary Fig. 8 (line 273-295).

5. Fig. 5a-5e, deficiency of MIB2 causes TNF α -induced apoptosis due to enhanced RIPK1 kinase activity has been reported earlier (Feltham et al., 2018). It is critical to discriminate the relationship between MIB2-mediated ubiquitination of cFLIPL and MIB2-mediated suppression of RIPK1 kinase activity. Therefore, the authors should use RIPK1 kinase inhibitor or genetically knockout RIPK1 in MIB2-KO cells to determine the contribution of ubiquitination of cFLIPL to the apoptosis in MIB2-KO cells. In addition, the authors need to examine the RIPK1 kinase activity in cFLIPL-3A-reconstituted MEFs to determine whether MIB2-mediated ubiquitination of cFLIPL could suppress RIPK1 kinase activation.

RESPONSE: Thank you for your thoughtful suggestion. We stimulated WT and *MIB2* KO cells with TNF/BV6 (0.1 μ M) in the absence or presence of Nec-1s. To our surprise, Nec-1s only partially blocked TNF/BV6-induced apoptosis in *MIB2* KO HeLa cells and did not block in *MIB2* KO HCT116 cells (Fig. 6g, h). Moreover, TNF/CHX-induced cell death was enhanced in *MIB2* KO HeLa cells, especially at low concentrations of TNF (Fig. 6i, j), whereas these cell death enhancements were not suppressed in the presence of Nec-1s. Consistently, we found that TNF/CHX did not induce phosphorylation of RIPK1 in WT and *MIB2* KO HeLa and HCT116 cells (Fig. 6k). These results suggest that MIB2 blocked, at least in part, RIPK1 kinase activity-independent cell death (line 252-257).

To further substantiate a role for ubiquitylation of cFLIP_L in suppression of TNF-induced cell death, we also generated a cFLIP_L mutant where MIB2-dependent ubiquitylated lysine residues were mutated to arginines (cFLIP_L-K9R). We then stimulated *Cflar*^{-/-} MEFs reconstituted with WT cFLIP_L, cFLIP_L-3A, or cFLIP_L-K9R with TNF and low concentrations of BV6 (0.1 μ M). TNF/BV6 did induce cell death in *Cflar*^{-/-} cFLIP_L-3A and cFLIP_L-K9R cells, but not WT cFLIP_L cells (Fig. 9d, h). Notably, TNF/BV6 did not induce RIPK1

phosphorylation in either transfectant (Fig. 9i), and RIPK1 inhibitor did not block TNF/BV6-induced cell death in *Cflar*^{-/-} cFLIP_L-3A and cFLIP_L-K9R cells (Fig. 9j). Together, MIB2 blocks RIPK1 kinase activity-independent cell death through ubiquitylation of cFLIP_L. We mentioned these points in the text (line 341-378), and made new Figures 6 and 9 to include these results.

6. Fig. 6c, 6f, although reconstitution of MIB2 KO cells with MIB2-MT (Ligase activity defective) was comparable to the apoptosis in MIB2 KO cells, it is not convincing to conclude defective ubiquitination of cFLIP_L mediated by MIB2 is required for TNF α -induced apoptosis in MIB2 KO cells, since MIB2 could also ubiquitinate RIPK1. More experiments are needed to determine whether RIPK1 kinase-dependent cell death is dominant in MIB2-MT reconstitution cells. First, the authors should examine whether MIB2-MT can still interact and ubiquitinate RIPK1. Second, the cell death assay should add a control using RIPK1 kinase inhibitor (eg, Necrostatin-1) to examine the contribution of RIPK1 kinase activity in the cell death of MIB2-MT reconstitution cells.

RESPONSE: Thank you for the thoughtful suggestion. Regarding the contribution of MIB2 to ubiquitylation of RIPK1, our data did not support the previous study (Feltham et al., 2018). First, ubiquitylation of RIPK1 in complex 1 in *MIB2* KO cells was comparable to the levels in WT cells after TNF stimulation (Fig. 5a and Supplementary Fig. 5b). Second, in the presence of IAP inhibitor that depletes cIAP1/2, ubiquitylation of RIPK1 in complex 1 was greatly reduced in both WT and *MIB2* KO cells (Fig. 6f and Supplementary Fig. 6), indicating that cIAP1/2 play a major role in ubiquitylation of RIPK1. Third, MIB2 constitutively bound RIPK1, and RIPK1 in the MIB2-containing complex was not ubiquitylated before TNF stimulation but underwent rapid ubiquitylation in *MIB2* KO HA-MIB2 WT HeLa cells 10 min after TNF stimulation (Fig. 10b). Indeed, we cannot formally exclude the possibility that contribution of MIB2 to RIPK1 ubiquitylation is cell-type specific. We made new Figures to include these results and mentioned it in the text (line 213-226, 241-251, 390-401).

To test whether HA-MIB2 MT interacts with RIPK1, we immunoprecipitated MIB2 with anti-HA antibody from *MIB2* KO cells reconstituted with HA-MIB2 WT or HA-MIB2 MT. Then, immunoprecipitates were analyzed by anti-RIPK1 antibody. As shown in Fig. 10c, MIB2 MT as well

as MIB2 WT constitutively bound RIPK1. Moreover, ubiquitylation of RIPK1 in the RIPK1/MIB2 complex in *MIB2* KO HeLa/HA-MIB2 MT cells was comparable to the levels in *MIB2* KO HeLa/HA-MIB2 WT cells, indicating that ubiquitin ligase activity is dispensable for RIPK1 ubiquitylation (Fig. 10c) (line 390-401).

Regarding contribution of RIPK1 kinase activity to enhanced cell death in *MIB2* KO cells, Nec-1s did not block TNF/CHX-induced apoptosis in *MIB2* KO HeLa/HA-MIB2 MT cells or *MIB2* KO HCT116/HA-MIB2 MT cells. Along this line, Nec-1s did not block TNF/BV6 (0.1 μ M)-induced cell death in *MIB2* KO HCT116-MIB2 MT cells. Nec-1s only partially blocked TNF/BV6 (0.1 μ M)-induced cell death in *MIB2* KO HeLa/HA-MIB2 MT cells, indicating that MIB2 suppresses RIPK1 kinase activity-dependent and -independent cell death. To include these results, We made new Figures 6 and 8 and mentioned these results in the text (line 228-268, 316-329).

7. Fig. 6h, the cFLIP_L level in the immunoprecipitates was missing; and an IgG control should be added to exclude the non-specific binding.

RESPONSE: Thank you for pointing out a critical issue. As suggested, we included amounts of cFLIP_L in the immunoprecipitates throughout the experiments. We detected more than three bands of cFLIP_L in the immunoprecipitates. To verify that these bands are indeed derived from cFLIP_L products, we tested whether knockdown of cFLIP_L by siRNA abolished these bands in the immunoprecipitates. As shown in Supplementary Fig. 1, all bands disappeared in cFLIP knockdown HeLa cells, indicating that these bands were indeed cFLIP_L. Since the molecular size of cFLIP_L was identical to the second band, the upper band may be modified cFLIP_L. The lower multiple bands indicated by asterisks may be degradation of products of cFLIP_L. The detailed characterization of the upper band of cFLIP_L will be investigated in the future study. We made a new Supplementary Fig. 1 and mentioned it in the text (line 110-117).

As suggested, we repeated similar experiments to include the results immunoprecipitated with control IgGs. We made a new Supplementary Fig. 10a to include the results.

8. Extended Fig. 5a, it is weird that IP-C, cFLIPL blotting has IgGs, but in

IP-H, cFLIPL blotting nearly has no IgGs.

Response: Sorry for confusing you. We used mouse IgGs (for control IP) and anti-HA (3F10 from rat) antibody for immunoprecipitation. Since the immunoprecipitates were blotted with anti-cFLIP_L antibody (7F10 from mouse) followed by HRP-conjugated anti-mouse IgGs antibody, heavy chains of mouse IgGs (lane C), but not rat IgG (lane H) exhibited strong signals. To avoid confusion, we repeated similar experiments using control rat IgGs and anti-HA antibody for immunoprecipitation. Now, the results clearly showed that anti-HA antibody, but not control IgGs, immunoprecipitated ubiquitylated cFLIP_L before TNF stimulation, but not after TNF stimulation. We replaced old results with new ones and included them in Fig. 10a.

9. Line 367, a spelling error “complex”

Response: As suggested, we corrected the typo.

10. Line 342-343, although TNF plus CHX is initially thought to induce RIPK1-independent cell death. However, under some conditions such as TBK1 deficiency, it could also induce RIPK1 kinase-dependent cell death (Xu et al., 2018). Since MIB2 could suppress RIPK1 kinase activation, TNF plus CHX may also induce RIPK1 kinase-dependent cell death in MIB2 KO cells. The authors should examine RIPK1 autophosphorylation and use RIPK1 kinase inhibitor to do the TNF plus CHX stimulation in MIB2 KO cells.

RESPONSE: Thank you for your thoughtful comments. We stimulated WT and *MIB2* KO HeLa cells with TNF/CHX in the absence or presence of Nec-1s. TNF/CHX-induced apoptosis was dramatically enhanced in *MIB2* KO HeLa and *MIB2* KO HCT116 cells compared to respective control wild-type cells when cells were stimulated with low concentrations of TNF (1~ 3 ng/ml). Nec-1s did not block TNF/CHX-induced apoptosis in *MIB2* KO cells at any concentrations of TNF (Fig. 6i, j). Moreover, we found that TNF/CHX stimulation did not induce RIPK1 phosphorylation in either WT or *MIB2* KO cells (Fig. 6k). Thus, we conclude that MIB2 suppresses RIPK1 kinase activity-independent cell death. We made a new Figure 6 to include these results and mentioned them in the text and (line 258-268).

11. Line 378-382, K63-linked ubiquitination of RIPK1 has been reported to suppress RIPK1 kinase activity by recruiting TAK1 kinase (Tang et al., 2019). The authors should examine the TAK1, IKK, TBK1, and MK2 recruitment in TNFR1 complex I in MIB2-KO cells to determine the exact molecular mechanism by which MIB2-mediated ubiquitination suppresses RIPK1 kinase activity.

RESPONSE: As suggested, we tested whether these kinases were recruited to the TNFR1 complex I. Consistent with a previous study (Lafont et al., 2018), TAK1, IKK, and TBK1 were recruited to the TNFR1 complex I in WT cells after TNF stimulation (Fig. 5a, Supplementary Fig. 5b). Notably, recruitment of these kinases to complex I in *MIB2* KO cells after TNF stimulation was comparable to the levels in WT cells. However, MK2 was not recruited to the TNFR1 complex I in either WT or *MIB2* KO cells. In contrast to the results in a previous study (Feltham et al., 2018), TNF-induced ubiquitylation of RIPK1 in *MIB2* KO cells was comparable to the levels in WT cells (Fig. 5a, Supplementary Fig. 5b).

MIB2 MT as well as MIB2 WT constitutively bound RIPK1.

Ubiquitylation of RIPK1 in the RIPK1/MIB2 complex in *MIB2* KO HeLa/HA-MIB2 MT cells was comparable to the levels in *MIB2* KO HeLa/HA-MIB2 WT cells, indicating that ubiquitin ligase activity of MIB2 is dispensable for RIPK1 ubiquitylation (Fig. 10c). In contrast, TNF/BV6 induced RIPK1 phosphorylation in *MIB2* KO HeLa cells, *MIB2* KO HeLa/HA-MIB2 MT, but not HA-MIB2 WT cells (Fig. 10d). Thus, ubiquitin ligase activity of MIB2 is dispensable for RIPK1 ubiquitylation, but indispensable for suppression of RIPK1 kinase activity. We made new Figures to include these results and mentioned them and discussed the mechanisms in the text (line 213-226, 390-401, 490-506) .

Supplementary References

- Dillon, C.P., A. Oberst, R. Weinlich, L.J. Janke, T.B. Kang, T. Ben-Moshe, T.W. Mak, D. Wallach, and D.R. Green. 2012. Survival function of the FADD-CASPASE-8-cFLIP(L) complex. *Cell Rep* 1:401-407.
- Feltham, R., K. Jamal, T. Tenev, G. Liccardi, I. Jaco, C.M. Domingues, O. Morris, S.W. John, A. Annibaldi, M. Widya, C.J. Kearney, D. Clancy, P.R. Elliott, T. Glatter, Q. Qiao, A.J. Thompson, A. Nesvizhskii, A. Schmidt, D. Komander, H. Wu, S. Martin, and P. Meier. 2018. Mind Bomb Regulates Cell Death during TNF Signaling by Suppressing RIPK1's Cytotoxic Potential. *Cell Rep* 23:470-484.
- Hillert, L.K., N.V. Ivanisenko, J. Espe, C. Konig, V.A. Ivanisenko, T. Kahne, and I.N. Lavrik. 2020. Long and short isoforms of c-FLIP act as control checkpoints of DED filament assembly. *Oncogene* 39:1756-1772.
- Hughes, M.A., I.R. Powley, R. Jukes-Jones, S. Horn, M. Feoktistova, L. Fairall, J.W. Schwabe, M. Leverkus, K. Cain, and M. MacFarlane. 2016. Co-operative and Hierarchical Binding of c-FLIP and Caspase-8: A Unified Model Defines How c-FLIP Isoforms Differentially Control Cell Fate. *Mol Cell* 61:834-849.
- Lafont, E., P. Draber, E. Rieser, M. Reichert, S. Kupka, D. de Miguel, H. Draberova, A. von Massenhausen, A. Bhamra, S. Henderson, K. Wojdyla, A. Chalk, S. Surinova, A. Linkermann, and H. Walczak. 2018. TBK1 and IKKepsilon prevent TNF-induced cell death by RIPK1 phosphorylation. *Nat Cell Biol* 20:1389-1399.
- Oberst, A., C.P. Dillon, R. Weinlich, L.L. McCormick, P. Fitzgerald, C. Pop, R. Hakem, G.S. Salvesen, and D.R. Green. 2011. Catalytic activity of the caspase-8-FLIP(L) complex inhibits RIPK3-dependent necrosis. *Nature* 471:363-367.
- Panayotova-Dimitrova, D., M. Feoktistova, M. Ploesser, B. Kellert, M. Hupe, S. Horn, R. Makarov, F. Jensen, S. Porubsky, A. Schmieder, A.C. Zenclussen, A. Marx, A. Kerstan, P. Geserick, Y.W. He, and M. Leverkus. 2013. cFLIP Regulates Skin Homeostasis and Protects against TNF-Induced Keratinocyte Apoptosis. *Cell Rep* 5:397-408.
- Piao, X., S. Komazawa-Sakon, T. Nishina, M. Koike, J.H. Piao, H. Ehlken, H. Kurihara, M. Hara, N. Van Rooijen, G. Schutz, M. Ohmuraya, Y. Uchiyama, H. Yagita, K. Okumura, Y.W. He, and H. Nakano. 2012. c-FLIP Maintains Tissue Homeostasis by Preventing Apoptosis and Programmed Necrosis. *Science signaling* 5:ra93.
- Shindo, R., M. Ohmuraya, S. Komazawa-Sakon, S. Miyake, Y. Deguchi, S. Yamazaki, T. Nishina, T. Yoshimoto, S. Kakuta, M. Koike, Y. Uchiyama, H. Konishi, H. Kiyama, T. Mikami, K. Moriwaki, K. Araki, and H. Nakano. 2019. Necroptosis

- of Intestinal Epithelial Cells Induces Type 3 Innate Lymphoid Cell-Dependent Lethal Ileitis. *iScience* 15:536-551.
- Shu, H.B., D.R. Halpin, and D.V. Goeddel. 1997. Casper is a FADD- and caspase-related inducer of apoptosis. *Immunity* 6:751-763.
- Thome, M., P. Schneider, K. Hofmann, H. Fickenscher, E. Meinel, F. Neipel, C. Mattmann, K. Burns, J.L. Bodmer, M. Schroter, C. Scaffidi, P.H. Krammer, M.E. Peter, and J. Tschopp. 1997. Viral FLICE-inhibitory proteins (FLIPs) prevent apoptosis induced by death receptors. *Nature* 386:517-521.
- Yeh, W.C., A. Itie, A.J. Elia, M. Ng, H.B. Shu, A. Wakeham, C. Mirtsos, N. Suzuki, M. Bonnard, D.V. Goeddel, and T.W. Mak. 2000. Requirement for Casper (c-FLIP) in regulation of death receptor-induced apoptosis and embryonic development. *Immunity* 12:633-642.
- Yu, J.W., P.D. Jeffrey, and Y. Shi. 2009. Mechanism of procaspase-8 activation by c-FLIPL. *Proc Natl Acad Sci U S A* 106:8169-8174.

Correlation of the revised Figures and initial Figures

Revised version	initial version
Fig. 1a-f	Fig. 1a-f
Fig. 2a-g	Fig. 2a-g
Fig. 3a-d	Fig. 3a, b, d, e
Fig. 4a-e	Fig. 4a-e
Fig. 4f	Fig. 3c
Fig. 4g, new results	
Fig. 5a	Fig. 7b + new results
Fig. 5b	Fig. 7a
Fig. 6a	Fig. 5a
Fig. 6b	Fig. 5d
Fig. 6c, new results	
Fig. 6d	Fig. 5a
Fig. 6e	Fig. 5d
Fig. 6f	Fig. 7e
Fig. 6g, new results	
Fig. 6h, new results	
Fig. 6i, new results	
Fig. 6j, new results	
Fig. 6k, new results	
Fig. 7a	Fig. 7c
Fig. 7b	Fig. 7d
Fig. 8a	Fig. 6a
Fig. 8b	Fig. 6d
Fig. 8c	Fig. 6c
Fig. 8d, new results	
Fig. 8e, new results	
Fig. 8f	Fig. 6g
Fig. 8g	Fig. 6f
Fig. 8h, new results	
Fig. 8i, new results	
Fig. 9a	Fig. 5f
Fig. 9b, new results	
Fig. 9c	Fig. 5g
Fig. 9d	Fig. 5h
Fig. 9e, new results	
Fig. 9f, new results	
Fig. 9g, new results	
Fig. 9h, new results	
Fig. 9i, new results	

Fig. 10a
Fig. 10b
Fig. 10c
Fig. 10d, new results
Fig. 10e, a new model
Fig. 10f, a new model

Supplementary Fig. 5a
Supplementary Fig. 5b
Supplementary Fig. 5a + new results

Revised version

initial version

Supplementary Fig. 1 a, b, new results

Supplementary Fig. 2a, b

Supplementary Fig. 1a, b

Supplementary Fig. 3a, b
Supplementary Fig. 3c

Supplementary Fig. 2a, b
Supplementary Fig. 2c + new results

Supplementary Fig. 4a, b, new information

Supplementary Fig. 5a
Supplementary Fig. 5b
Supplementary Fig. 5c

Fig. 5c
Supplementary Fig. 4b + new results
Supplementary Fig. 4a

Supplementary Fig. 6, new results

Supplementary Fig. 7a-d, new results

Supplementary Fig. 6a-d

Supplementary Fig. 8a-h, new results

Supplementary Fig. 3a-d

Supplementary Fig. 9a, b, new results

Supplementary Fig. 4c, d

Supplementary Fig. 10a, b, new results

Fig.6h, i

Reviewers' comments:

Reviewer #1 (Remarks to the Author):

The authors have addressed most of my comments with regard of c-FLIP isoforms, however, I am still not convinced that apoptosis acceleration upon MIND2 KO in TNF/BV6 co-stimulation is strongly dependent on the ubiquitination of c-FLIP at the C-terminal domain.

The additional data that were provided in the revised version along with answers strongly support that RIPK1 and MIN2 interactions is a key that defines life and death decisions in the cell. Indeed, it is the level of RIPK1 ubiquitination in the cells that strongly defines the amount of complex II formation, which in turn is defined by the activity of MIN2. This is clearly observed by all co-immunoprecipitations of complex II that are performed in the study. These immunoprecipitations show that upon MIND2 KO, there is much more RIPK1 co-immunoprecipitated in the complex II. This certainly has a primary influence on the amount of complex II rather than the ubiquitination of c-FLIP. Indeed, the complex II formation is based on the interactions of DD of FADD and RIPK1. Subsequently, the observed absence of RIPK1 in FADD-IP or decreased amounts of pRIPK1 indicate that there is a mechanism present in the cells that inhibits RIPK1 from the efficient formation of this complex (Figures 6, and 7).

I would fully support the point of view that ubiquitinated c-FLIP might further attenuate caspase-8 activity in this complex, but I can not support the current scheme suggested in the Figure 10, that ubiquitinated c-FLIP disrupts the formation of complex II. This, by the way, would preclude any complex II formation and apoptosis in the cells that have MIND2, which is not the case.

Hence, I would suggest to largely rewrite the manuscript pointing out on the double role of MIN2 both on RIPK1 as a primary signal for complex II formation and then on c-FLIP as an attenuating signal and taking away the suggested mechanism of disrupting DED filaments by c-FLIP ubiquitination.

Moreover, the destruction of the DED Filaments by c-FLIPL ubiquitination is also not possible simply due to the quantitative proteomics Analysis, which have shown that c-FLIPL is substoichiometric in the Filament, in particular, in HeLa cells. This comment goes along with my previous comment on the low numbers of c-FLIPs in HeLa cells.

Moreover, along the lane 509, the authors write that, intriguingly, a previous study reported that low concentrations of cFLIPL promote caspase 8 activation in a cell-free system, whereas high concentrations of cFLIPL suppress caspase 8 activation 26. Low concentration of c-FLIP promoting caspase-8 activation was also shown in the cellular systems, and not only using cell-free systems. The corresponding publications have to be cited in this regard:

<https://www.ncbi.nlm.nih.gov/pmc/articles/PMC125398/>

<https://pubmed.ncbi.nlm.nih.gov/12215447/>

<https://pubmed.ncbi.nlm.nih.gov/20696707/>

Finally, as for the mechanism of attenuation of caspase-8 activity I would suggest that indeed the conformational changes in the heterodimer procaspase-8-c-FLIPL upon ubiquitination of the

residues within the vicinity of 468 (Yu et al) might lead to the drastic changes in the activity of the heterodimer. This likely are the effects observed in Figures: 1, 2, 6, and 7.

Reviewer #2 (Remarks to the Author):

The authors have addressed the issues raised in a comprehensive manner. I would just request for the ease of the reader that they label their IP experiments consistently with the IP antibody used (as in Fig. 1c, 1d, 3b, 4). This is not done in Fig. 1e, 1f, 2, 3d, 8a, 9a, 9e, and 10a-c.

Reviewer #3 (Remarks to the Author):

The revised manuscript has adequately addressed the main concerns raised in the original submission with new, or improved, data, and consequently have strengthened the manuscript considerably.

Reviewers' comments:

Reviewer #1 (Remarks to the Author):

The authors have addressed most of my comments with regard of c-FLIP isoforms, however, I am still not convinced that apoptosis acceleration upon MIND2 KO in TNF/BV6 co-stimulation is strongly dependent on the ubiquitination of c-FLIP at the C-terminal domain.

The additional data that were provided in the revised version along with answers strongly support that RIPK1 and MIND2 interactions is a key that defines life and death decisions in the cell. Indeed, it is the level of RIPK1 ubiquitination in the cells that strongly defines the amount of complex II formation, which in turn is defined by the activity of MIND2. This is clearly observed by all co-immunoprecipitations of complex II that are performed in the study. These immunoprecipitations show that upon MIND2 KO, there is much more RIPK1 co-immunoprecipitated in the complex II. This certainly has a primary influence on the amount of complex II rather than the ubiquitination of c-FLIP. Indeed, the complex II formation is based on the interactions of DD of FADD and RIPK1. Subsequently, the observed absence of RIPK1 in FADD-IP or decreased amounts of pRIPK1 indicate that there is a mechanism present in the cells that inhibits RIPK1 from the efficient formation of this complex (Figures 6 and 7).

I would fully support the point of view that ubiquitinated c-FLIP might further attenuate caspase-8 activity in this complex, but I can not support the current scheme suggested in the Figure 10, that ubiquitinated c-FLIP disrupts the formation of complex II. This, by the way, would preclude any complex II formation and apoptosis in the cells that have MIND2, which is not the case.

Hence, I would suggest to largely rewrite the manuscript pointing out on the double role of MIND2 both on RIPK1 as a primary signal for complex II formation and then on c-FLIP as an attenuating signal and taking away the suggested mechanism of disrupting DED filaments by c-FLIP

ubiquitylation.

Moreover, the destruction of the DED Filaments by c-FLIPL ubiquitylation is also not possible simply due to the quantitative proteomics analysis, which have shown that c-FLIPL is substoichiometric in the Filament, in particular, in HeLa cells. This comment goes along with my previous comment on the low numbers of c-FLIPs in HeLa cells.

RESPONSE: Thank you for pointing out critical issues. Indeed, we did not directly show that ubiquitylated cFLIP_L disrupts the complex II formation. According to your suggestion, we have deleted several sentences showing that ubiquitylated cFLIP_L blocks high-ordered oligomer or complex II formation in the former manuscript (lines 210-211, 306-308, 507-509, 522-523). We only mentioned that ubiquitylated cFLIP_L interacts with caspase 8, but fails to activate caspase 8 (Fig. 10e). Moreover, to show a crucial role for MIB2-dependent suppression of RIPK1 kinase activity in attenuation of TNF-induced apoptosis, we have made a new Fig. 10e to include a model how MIB2 suppresses the complex IIb formation. We have mentioned these points in the Results, Discussion, and Figure legends (lines 307-310, 404-411, 507-508, 525-526, 528-529, 1094-1101).

Along with this line, we have changed the title to “MIND bomb 2 prevents RIPK1 kinase activity-dependent and -independent apoptosis through ubiquitylation of cFLIP_L”.

Moreover, along the lane 509, the authors write that, intriguingly, a previous study reported that low concentrations of cFLIPL promote caspase 8 activation in a cell-free system, whereas high concentrations of cFLIPL suppress caspase 8 activation 26. Low concentration of c-FLIP promoting caspase-8 activation was also shown in the cellular systems, and not only using cell-free systems. The corresponding publications have to be cited in this regard:

<https://www.ncbi.nlm.nih.gov/pmc/articles/PMC125398/>

<https://pubmed.ncbi.nlm.nih.gov/12215447/>

<https://pubmed.ncbi.nlm.nih.gov/20696707/>

RESPONSE: Thank you for the thoughtful suggestion. As suggested, we have included the references and mentioned these in the text (lines 511-514, 516-518).

Finally, as for the mechanism of attenuation of caspase-8 activity I would suggest that indeed the conformational changes in the heterodimer procaspase-8-c-FLIPL upon ubiquitination of the residues within the vicinity of 468 (Yu et al) might lead to the drastic changes in the activity of the heterodimer. This likely are the effects observed in Figures: 1, 2, 6, and 7.

RESPONSE: Thank you for pointing out a critical issue. Indeed, the complex II formation was greatly facilitated in *MIB2* KO cells, suggesting that the anti-apoptotic function of MIB2 might be primarily mediated by suppression of the complex II formation. However, we found that RIPK1 kinase activity-independent cell death was enhanced in *MIB2* KO cells (Fig. 6g, h, i, j) and cells lacking MIB2's ubiquitin ligase activity (Fig. 8d, e, h, i). Moreover, as shown in Fig. 9, TNF/BV6-induced apoptosis was enhanced in cells expressing cFLIP_L mutants lacking the binding domain of MIB2 or MIB2-dependent ubiquitylation sites (Fig. 9d, h, j). These cell death enhancement might be caused by a decrease in ubiquitylation of cFLIP_L, but not modulating RIPK1 kinase activity. Thus, we assume that MIB2-dependent ubiquitylation of cFLIP_L contributes, at least in part, to attenuation of TNF-induced apoptosis.

References

Chang, D.W., Z. Xing, Y. Pan, A. Algeciras-Schimnich, B.C. Barnhart, S. Yaish-Ohad, M.E. Peter, and X. Yang. 2002. c-FLIP(L) is a dual function regulator for caspase-8 activation and CD95-mediated apoptosis. *EMBO J* 21:3704-3714.

Fricker, N., J. Beaudouin, P. Richter, R. Eils, P.H. Krammer, and I.N. Lavrik. 2010. Model-based dissection of CD95 signaling dynamics reveals both a pro- and antiapoptotic role of c-FLIPL. *J Cell Biol* 190:377-389.

Micheau, O., M. Thome, P. Schneider, N. Holler, J. Tschoop, D.W. Nicholson, C. Briand, and M.G. Grutter. 2002. The long form of FLIP is an

activator of caspase-8 at the Fas death-inducing signaling complex. *J Biol Chem* 277:45162-45171.

Reviewer #2 (Remarks to the Author):

The authors have addressed the issues raised in a comprehensive manner. I would just request for the ease of the reader that they label their IP experiments consistently with the IP antibody used (as in Fig. 1c, 1d, 3b, 4). This is not done in Fig. 1e, 1f, 2, 3d, 8a, 9a, 9e, and 10a-c.

RESPONSE: Thank you for your appreciation of our revised manuscript. As suggested, we described the IP antibodies in the Figures.

Reviewer #3 (Remarks to the Author):

The revised manuscript has adequately addressed the main concerns raised in the original submission with new, or improved, data, and consequently have strengthened the manuscript considerably.

RESPONSE: Thank you for your appreciation of our revised manuscript.

REVIEWERS' COMMENTS:

Reviewer #1 (Remarks to the Author):

my comments were addresssed